neuroscience, behaviour, ecology

visual detection, sit-and-wait predator, breaking camouflage, active sensing, eyeshine, cryptobenthic fish

**Author for correspondence:**
Nico K. Michiels
e-mail: nico.michiels@uni-tuebingen.de

# Redirection of ambient light improves predator detection in a diurnal fish

Matteo Santon[1], Pierre-Paul Bitton[1,3], Jasha Dehm[1,2], Roland Fritsch[1], Ulrike K. Harant[1], Nils Anthes[1] and Nico K. Michiels[1]

[1]Animal Evolutionary Ecology, Institute of Evolution and Ecology, Department of Biology, Faculty of Science, University of Tübingen, Auf der Morgenstelle 28, 72076 Tübingen, Germany
[2]School of Marine Studies, Faculty of Science, Technology and Environment, University of the South Pacific, Laucala Bay Rd, Suva, Fiji
[3]Department of Psychology, Memorial University of Newfoundland, 232 Elizabeth Avenue, St John's, NL Canada, A1B 3X9

MS, 0000-0002-9397-4052; NKM, 0000-0001-9873-3111

Cases where animals use controlled illumination to improve vision are rare and thus far limited to chemiluminescence, which only functions in darkness. This constraint was recently relaxed by studies on *Tripterygion delaisi*, a small triplefin that redirects sunlight instead. By reflecting light sideways with its iris, it has been suggested to induce and detect eyeshine in nearby micro-prey. Here, we test whether 'diurnal active photolocation' also improves *T. delaisi*'s ability to detect the cryptobenthic sit-and-wait predator *Scorpaena porcus*, a scorpionfish with strong daytime retroreflective eyeshine. Three independent experiments revealed that triplefins in which light redirection was artificially suppressed approached scorpionfish significantly closer than two control treatments before moving away to a safer distance. Visual modelling confirmed that ocular light redirection by a triplefin is sufficiently strong to generate a luminance increase in scorpionfish eyeshine that can be perceived by the triplefin over 6–8 cm under average conditions. These distances coincide well with the closest approaches observed. We conclude that light redirection by small, diurnal fish significantly contributes to their ability to visually detect cryptic predators, strongly widening the conditions under which active sensing with light is feasible. We discuss the consequences for fish eye evolution.

## 1. Introduction

Vision represents one of the most extraordinary outcomes of natural selection [1], as exemplified by studies across the animal kingdom that describe how visual systems are adapted to a species' ecology [2,3]. Given how diverse such systems can be, it is surprising that adaptations to improve vision by means of controlled illumination are rare [4]. A well-known exception are nocturnal flashlight fishes. They feature a subocular chemiluminescent light organ just below the pupil [5] that facilitates schooling behaviour at night [6] and glows sufficiently strong to illuminate and detect nearby prey [7]. By being next to the visual axis, it is ideally positioned to induce and detect retroreflective eyeshine (cat's eyes) in nearby organisms [8]. This follows from the fact that retroreflective eyes return incoming light back to the source in a narrow beam, regardless of the incoming angle [9,10]. This places them among the strongest and most easily detected biological reflectors—provided the observer has a source of light next to the pupil [11].

Recent findings in the triplefin *Tripterygion delaisi* suggest that diurnal fish use an analogous principle by actively redirecting downwelling sunlight using the lower iris, generating a so-called ocular spark (figure 1a; electronic supplementary material, figure S4) [13]. Ocular sparks arise because the lens

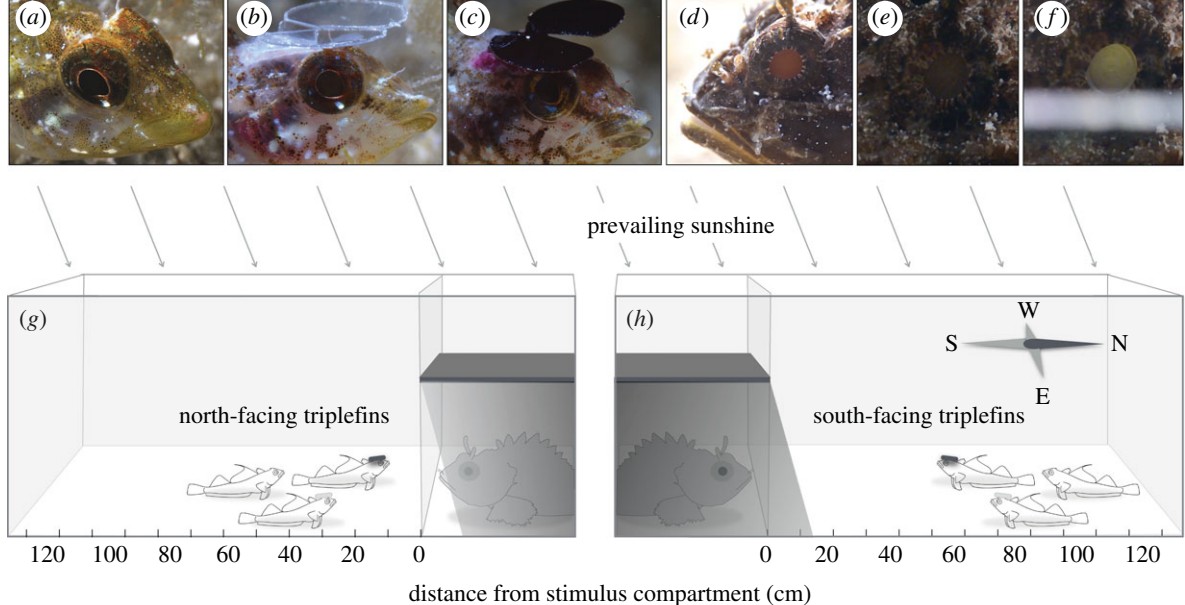

**Figure 1.** Experimental design to test for diurnal active photolocation in the bottom-dwelling triplefin *Tripterygion delaisi*. Three treatments were tested (*a*) unhatted sham control, (*b*) clear-hatted control and (*c*) shading hat treatment. While (*a*) and (*b*) can re-direct light using a blue ocular spark (spot of focused light on the lower iris), (*c*) cannot. (*d*) The scorpionfish *Scorpaena porcus* shows daytime eyeshine [12]. (*e,f*) Coaxial 'illumination' with a strip of white paper reveals that the eyeshine has a strong retroreflective component (taken from https://doi.org/10.6084/m9.figshare.5902720.v1). (*g,h*) Triplets of one triplefin per hat treatment were exposed to a shaded predator (as shown) or stone behind a windowpane. In the field, we tested two orientations (triplefins facing north or south, as shown). The response variable was distance from the stimulus compartment. Drawings not to scale (see Material and methods). Pictures by M.S. and N.K.M. See electronic supplementary material, figure S1 for additional pictures of triplefins and scorpionfish in the two field set-ups. (Online version in colour.)

protrudes from the pupil, allowing some of the downwelling light to pass through the lens without entering the eye, and be focused in a bright spot on the iris below. As a result, sunlight is reflected sideways, outside the approximately 97° angular range of downwelling light dictated by Snell's window [14]. A recently published model confirmed that ocular sparks allow *T. delaisi* to induce perceptible eyeshine in micro-prey (gammarid shrimps) under biologically plausible scenarios [15]. We define this as 'diurnal active photolocation', a process by which ocular radiance improves visual detection of nearby cryptic organisms. This new form of active sensing was originally proposed by Collin B. Jack [11] and can be placed between 'teleceptive active sensing' (perception of the reflections of a self-produced signal) and 'contact active sensing' (touch perception) [4].

In this study, we test whether ocular sparks also improve a triplefin's ability to detect the black scorpionfish *Scorpaena porcus*, a cryptic, sit-and-wait predator with large, retroreflective eyes [16–18]. Whereas retroreflective eyes are mainly known as an adaptation to vision in dim light [19], daytime eyeshine in cryptobenthic predatory fish improves their camouflage by concealing their black pupils. This has already been shown for *S. porcus* [12] (figure 1*d–f*). We present data from three experiments in which we manipulated ocular spark production in triplefins by attaching opaque mini-hats (figure 1*c*). We compared their response with two control treatments (figure 1*a,b*) to a scorpionfish behind a windowpane (figure 1*g,h*) or a stone as a control visual stimulus. We predicted control-treated triplefins to keep a greater safe distance from scorpionfish than shading-hatted triplefins. To validate the empirical results, we subsequently implemented visual models to assess if and from which distance a perceptible change in contrast in the pupil of a scorpionfish can be induced by a triplefin when producing an ocular spark under natural conditions.

## 2. Methods

### (a) Model species and location

Triplefins (family Tripterygiidae) are small, cryptobenthic micro-predators on marine hard substrates. With a standard length of 3–5 cm *Tripterygion delaisi* (electronic supplementary material, figure S1) is one of the larger species. It occurs in the NE Atlantic and Mediterranean between 3 and 50 m depth, with highest densities in 5–15 m. Except for breeding males, it is highly cryptic. Unlike other small benthic fish, triplefins do not have a place to retreat to [20]. Instead, they forage for micro-prey using saltatory movement patterns [21] in which short hops are alternated with prolonged periods of assessing the surroundings. They possess high-amplitude, independent eye movement, and high contrast sensitivity and visual acuity for their size [22,23].

The scorpionfish *Scorpaena porcus* (family Scorpaenidae) is a cryptobenthic sit-and-wait predator (12–20 cm) from coastal marine hard substrates and seagrass in the NE Atlantic and Mediterranean Sea [24] (electronic supplementary material, figure S1). It feeds on crustaceans and small benthic fish [25]. Scorpionfish eyes possess a reflective *stratum argenteum* and a translucent retinal pigment epithelium both of which contribute to daytime eyeshine [12].

Fish and data were collected at STARESO (Station de Recherches Sous Marines et Océanographiques, Calvi, France) under the permit of the station. The hatting technique was developed at the University of Tübingen, permit ZO1–16 Regierungspräsidium Tübingen. We caught fish by hand-netting while SCUBA diving and stored them in large tanks with running fresh seawater. They were returned to the field afterwards.

### (b) Blocking ocular sparks with mini-hats

We prevented ocular spark formation using mini shading hats excised from a dark red polyester filter (transmission 1%, LEE #787 'Marius Red', LEE Filters). Clear filter hats (LEE #130, 'Clear') represented a control for the shading. No hat, but an otherwise identical procedure served as control for the presence

of a hat. Hats had a triangular base for attachment and two raised, forward-projecting wings, assuring free eye movement with an unobstructed forward view (greater than 45° from horizontal; figure 1b,c). Hats varied in diameter (6–9 mm) to match head size. Clear-hatted and unhatted triplefins regularly generated ocular sparks (figure 1a,b).

For hatting, we anaesthetized fish (100 mg l$^{-1}$ MS-222 in seawater, pH = 8.2) until all movements ceased except for breathing (3–4.5 min). The dorsal head surface was dried with paper tissue. Hats were glued to the dorso-posterior head area using surgical glue (Surgibond, Sutures Limited, UK or Vetbond Tissue Adhesive, 3 M). After polymerization for 45 s, fish were moved to recovery containers with aerated seawater and regained consciousness within 5–10 min. Survival rate was 97.4%. Most hats detached spontaneously within 0–4 days. All fish were used only once. Pilot experiments confirmed that typical behaviours such as fin-flicks, bobbing, overall movement, and head and eye movements did not differ between treatments [26].

## (c) Laboratory experiment

We used four aquaria (L × W × D: 130 × 50 × 50 cm$^3$) with a white, barren bottom (avoided by triplefins) and added a 10 cm wide strip of gravel along the long side as a preferred substrate. At one end, we placed two perforated containers with a glass front (L × W × H: 24 × 14 × 16 cm$^3$) one with a stone, one with a scorpionfish. Only one of the two was visible to the triplefins on a given day. Each aquarium was illuminated with a 150 W cold white LED floodlight (TIROLED Hallenleuchte, 150 W, 16 000 Lumen) shielded with a LEE Filters #172 Lagoon Blue filter. The containers with the stimuli were shaded. On day 1, triplefins were hatted or sham-treated and placed in the aquarium in the late afternoon. Data collection started the next day, approximately 14 h later to allow hatted fish to recover and explore the new environment. Observations took place on days 2 and 3. Two aquaria started with a scorpionfish as the visible stimulus, the others with a stone. Stimuli were swapped after day 2. We assessed the distance of each triplefin to the stimulus at 08.00, 11.00, 13.00, 15.00 and 18.00. Twenty triplets of size-matched T. delaisi were tested (n = 60). Owing to premature hat loss in five of these, 15 triplets were used for the analysis.

## (d) Replicate experiment in the field

We replicated the laboratory experiment using 10 tanks of Evotron Plexiglas (L × W × D: 150 × 25 × 50 cm$^3$) placed at 15 m depth on a sandy patch (figure 1g,h; electronic supplementary material, figure S1). To control for the position of the sun, five tanks faced north, five south. We used local bright sand mixed with gravel as substrate for the triplefin compartment (125 × 25 cm$^2$). This was separated by transparent Plexiglass from the shaded stimulus display compartment (15 × 25 cm$^2$). An additional, similar-sized area behind the latter was used to hide the alternative stimulus. All compartments had many slits and holes for water exchange to assure that the scorpionfish could be smelled and heard [27], even when only the stone was visible. Visual contact between fish and the outside was excluded by a 10 cm white side cover along the bottom edge. Distance markers at 1, 5 and 10 cm were present along both long sides of the enclosures. As the main response variable, we noted the distance of each individual from the stimulus by aligning the head of the triplefins with that of the distance markers on both sides while hovering 50 cm above the enclosure. This is possible because triplefins freeze rather than flee when large fish (or divers) pass. Distance data were collected at three times (09.00, 12.00 and 15.00) on 2 days following deployment in the afternoon of the day before. More time points were not possible because of SCUBA diving restrictions. Stimuli were swapped

at the end of the first observation day. Fifty triplets (n = 150 triplefins) were tested, 25 per orientation.

## (e) Immediate response to a scorpionfish in the field

In the preceding experiments, we concluded that recovery from anaesthesia is completed within 2–3 h, and that shorter tanks (50 cm) are sufficient to reveal a response. Further pilot trials showed that individuals are more likely to move when placed on dark sand, leading to a faster response. Based on these modifications, a new paradigm was implemented in the second field experiment (field 10 m, electronic supplementary material, figure S1). It allowed us to carry out a complete run in a single day, also reducing the problem of data loss due to premature hat loss. Here, we exposed single, shaded and clear-hatted triplefins to a shaded scorpionfish to test their immediate response in 90–100 min following release in the tank. As before, we used 10 Plexiglass tanks, five with triplefins facing north, another five facing south. Tank layout was identical (figure 1), but smaller, with 50 × 25 cm$^2$ for the triplefins and 12 × 25 cm$^2$ for the scorpionfish (electronic supplementary material, figure S1). To improve diving safety and standardize field conditions, tanks were mounted on stable floats 1.5 m above the seagrass at 10 m depth. The triplefins now moved on dark sand and we used black side covers to block their view to the outside, creating a slightly darker, more natural environment than the bright conditions of the previous field experiment. Triplefins were added at the 25 cm mark and its first position was determined approximately 1 min after release. Once a triplefin had been released in each tank, all tanks were visited another three times. After an approximately 30 min surface interval, the divers went back to collect another three data points before collecting all fish. This resulted in distance estimates for seven time points up to 100 min after initial release. Eight cohorts of 10 triplefins were observed, 38 shaded and 42 clear-hatted. Using controlled randomization, treatments were equally distributed across cohort, tank ID and orientation.

## (f) Statistical analysis
### (i) Repeatability analysis

In all experiments, distance measurements were not blind for hat treatment. However, room for error was limited as we did not interpret a behaviour, but noted the position of the head of a fish relative to a ruler placed alongside the tank. In the laboratory, observer, fish and ruler were close to each other and easy to align for virtually error-free measurements at the ±1 cm scale. In the field, a SCUBA diver was hovering above the tank and used rulers on both long sides for head alignment to determine fish position. To test repeatability in the field, both divers who collected the data (MS, UKH) determined 116 distances of triplefins in the first field experiment. Using the R package rptR [28], datatype Gaussian and 1000 permutations, the repeatability estimate was R = 0.995 (likelihood ratio test: p < 0.0001).

### (ii) Statistical model choice and pooling of controls

Distance from the display compartment was the response variable in two linear and one generalized linear mixed models (GLMMs) generated with the lme4 package [29] and glmmTMB package [30] for R v. 3.4.3. [31]. For the first two experiments, we first compared the two control treatments (sham and clear hat) to verify that carrying a hat did not affect behaviour, and to confirm their ability to distinguish a cryptic predator from a stone (see Results and electronic supplementary material, tables S1–S2). In the first field experiment, hat loss resulted in a considerable drop in sample size: only 22 out of the 50 triplets tested remained intact. To recover some of the lost data points, we used the similarity between clear-hatted and sham controls to average their

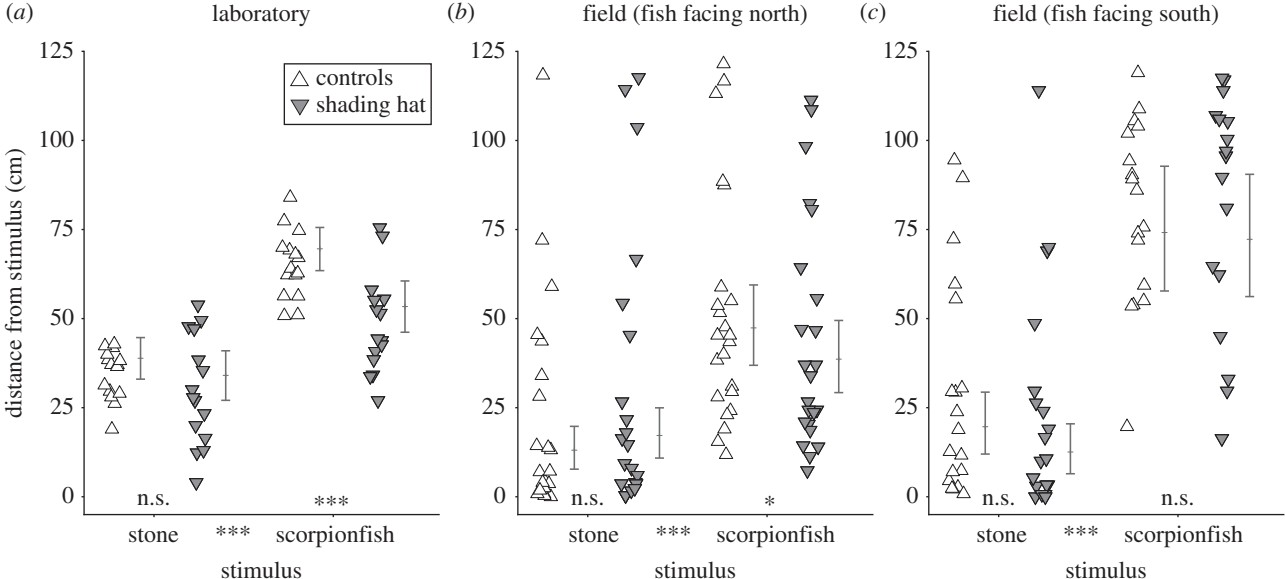

**Figure 2.** Effect of *hat treatment* (key predictor) on the average *distance* (response) from the stimulus as a function of the predictor *stimulus* (stone or scorpionfish). Data from two replicate experiments in the laboratory (*a*) and the field (*b–c*). (*a*) Shaded individuals stayed significantly closer to a scorpionfish than the controls (*n* = 15 triplets). (*b*) Among north-facing triplefins shaded individuals stayed closer to a scorpionfish than the controls (*n* = 24 triplets). (*c*) Among south-facing triplefins such an effect was absent, but all treatments responded more strongly to the scorpionfish (*n* = 19 triplets). Symbols: average per shaded individual or for both control individuals; error bars: model-predicted group means ± 95% credible intervals; \*\*\*$p < 0.001$, \*$p < 0.05$, n.s. $p > 0.05$. Note: statistical comparisons rested on connected measures *within* triplets and five (laboratory) and three (field) data points per stimulus, making error bars imprecise indicators of significance (electronic supplementary material, tables S1 and S2).

responses. This 'control average' was then used for the comparison with the shaded treatment in the final models. This allowed us to include triplets in which only the clear-hatted fish had lost its hat and had become visually indistinguishable from the hatless sham treatment. This procedure did not affect the direction of the results (see also electronic supplementary material, figure S2), yet increased sample sizes from 13 to 24 triplets in the north-facing field group, and from nine to 19 in the south-facing group. This includes triplets that were complete during the first observation day, but were excluded for the second day because the shaded individual had lost its hat after the first day. All this explains the variation in triplet numbers seen in the analyses.

### (iii) Predictors and transformations

For the laboratory experiment (figure 2*a*; electronic supplementary material, table S1), the initial fixed model component included the main predictors *stimulus* (stone versus scorpionfish), *hat treatment* (no hat versus clear hat, or averaged controls versus shaded) and their interaction. We further included the fixed covariates *time of day* for each observation, *stimulus order*, *cohort* and *tank ID*.

The models for the field replicate (figure 2*b*,*c*; electronic supplementary material, table S2) also included the fixed factor *orientation* (north or south), and its interactions with the main predictors. Both models were implemented using a normal distribution. However, we square-root-transformed the response variable *distance* to improve residual homogeneity in the analysis of the first field experiment. The transformation of the response variable did not cause any change in the effects of the interactions between covariates. Models to compare the response of controls versus shaded fish were calculated separately for north versus south orientations because fish responded differently to the scorpionfish depending on the random predictor *orientation* (electronic supplementary material, table S2*a*).

For the third experiment (figure 3; electronic supplementary material, table S3), the initial fixed model component included the main predictors *hat treatment* (clear hat or shaded), *time*,

*orientation*, and their three-way interaction. We also included *time* as a quadratic component to explain the nonlinear patterns of the data, assessed using the gam function of the mgvc R package [32], and the covariate *day*, as data were collected on three subsequent days. In this experiment, enclosures were shorter (50 cm) and distance data, therefore, more bounded and less normal. The response variable was transformed as proportion ($0 < x < 1$) of *distance* obtained by dividing all distances by the maximum length of the tank plus one (51 cm). The transformation of the response variable did not affect the interactions between covariates, yet allowed us to use a beta-binomial distribution, thus improving residual homogeneity. We finally included a first-order autoregressive (AR1) variance structure to correct for temporal dependency in the observations of the same individuals. See electronic supplementary material, Methods S1 for further details on random factors and model selection.

### (g) Visual modelling

To validate the results, we computed the contrast change in the pupil of a scorpionfish as perceived by an untreated triplefin when producing an ocular spark. We informed the model using previously published parameters [12,13,15,22,23] supplemented by additional measurements under the conditions of the last field experiment, including the baseline radiance of scorpionfish pupils [12]. We limited ourselves to modelling the effect of blue ocular sparks, which is the stronger of the two types known from *T. delaisi* [13]. See electronic supplementary material, table S4 for symbol definitions and electronic supplementary material, figures S3 and S4 for a visualization of the formulae used. Note that we express reflectance as a proportion (not %) to facilitate calculations.

### (i) Estimating scorpionfish pupil radiance with and without ocular spark

We assumed both triplefins and scorpionfish were looking orthogonally at one another to calculate the photon flux of the scorpionfish pupil reaching the triplefin pupil. Using retinal

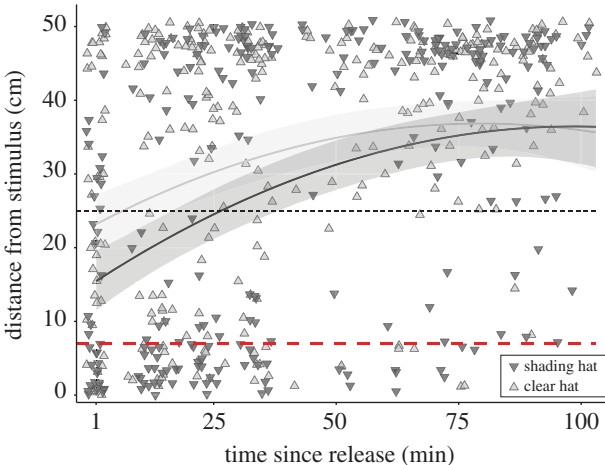

**Figure 3.** Distance from a scorpionfish as a function of hat treatment and time since release in the field at 10 m depth. Measurements ($n = 7$ per ind.) started 1 min after releasing a triplefin in the middle of a 50 cm long tank ($n$ clear hat = 42, $n$ shading hat = 38, electronic supplementary material, figure S1). Curves show predictions from a generalized linear mixed model for shaded (dark grey) and clear-hatted (light grey) triplefins including 95% credible intervals as shades (electronic supplementary material, table S3). Short-dashed line: point of release (approx. 25 cm). Long-dashed line: average distance of approximately 7 cm at which diurnal active photolocation is predicted to allow triplefins to induce and perceive scorpionfish eyeshine (range 3–14 cm, figure 4). Data points were slightly jittered to reveal overlapping observations. (Online version in colour.)

quantum catch estimates, we calculated the chromatic contrast [33] between the scorpionfish pupil with and without the contribution of the blue ocular sparks. The achromatic contrast between the same two conditions was estimated by calculating the Michelson contrast using the sum of the quantum catches of the two MWS/LWS double cone photoreceptors of *T. delaisi*. For comparison, we performed the same calculations using photon flux from the scorpionfish iris with and without the contribution of an ocular spark. We parameterized the equations using measurements of: (1) ambient light in the tanks at 10 m depth, (2) ocular spark radiance under downwelling light conditions, (3) baseline scorpionfish pupil radiance in the experimental tanks, (4) sizes of triplefin pupil, ocular spark and scorpionfish pupil and (5) scorpionfish iris and pupil reflectance. The latter was measured in a dark room using an ophthalmoscopic set-up [12]. In the same paper, retroreflective eyeshine is termed 'SAR narrow-sense eyeshine', the retroreflective component of the *stratum argenteum*-reflected eyeshine. A detailed description of how spectrometric measurements were taken and solid angles were estimated can be found in electronic supplementary material, methods S2.

### (ii) Visual models and maximum detection distance

The receptor-noise limited model for calculation of chromatic contrast was informed using triplefin ocular media transmission values, photoreceptor sensitivity curves [34,35], and the relative photoreceptor density of single to double cones of $1:4:4$ as found in the triplefin fovea [22]. We treated double cones as two independent cones [36]. We used a Weber fraction ($\omega$) value of 0.05 as in previous studies [37,38]. Chromatic contrasts are measured as just-noticeable differences (JNDs), where values greater than 1 are considered to be larger than the minimum discernible difference between two objects. We calculated the Michelson achromatic contrast as

$$C = \frac{(Q_1 - Q_2)}{(Q_1 + Q_2)}$$

where $Q_1$ and $Q_2$ are the sum of the quantum catches of the two members of the double cones which are associated with the achromatic channel, under photon flux$_1$ and photon flux$_2$. Flux$_1$ is the sum of the photon flux into a triplefin's eye caused by the baseline radiance of a scorpionfish pupil and the photon flux caused by the retroreflection of an ocular spark in the scorpionfish pupil (sum of the equations explained in electronic supplementary material, figures S3 and S4). Flux$_2$ is calculated from the baseline radiance of a scorpionfish pupil only (no ocular spark reflection, electronic supplementary material, figure S3). Both fluxes were calculated for each nanometre before calculating per-cone quantum catches over a 400–700 nm range. We determined the maximum discernible distance of the ocular spark radiance reflected through a scorpionfish pupil by calculating the chromatic and achromatic contrast at each millimetre, between 1 and 15 cm, and extracting the first value at which the contrast was equal to or exceeded the threshold of 1.0 JND for chromatic contrasts and 0.008 for achromatic Michelson contrasts as measured in *T. delaisi* [23] and other fish species [39]. All visual models were performed using the R package pavo [40].

## 3. Results

### (a) Long-term response of triplets (laboratory and first field experiment)

After overnight acclimatization, we observed triplefins at longer distances from the predator than from the stone, irrespective of hat treatment (all LMMs: *stimulus* $p < 0.0001$) (figure 2; electronic supplementary material, figure S1 and tables S1–S2). This effect was strongest in triplefins from the south-facing field tanks. All this indicates that triplefins can visually distinguish a stone from a scorpionfish independent of light redirection. Since the response of the two control treatments was indistinguishable (LMM: *hat treatment* $p_{lab}$ = 0.373, $p_{field}$ = 0.844, electronic supplementary material, tables S1a–S2a), we averaged the distances of the controls per triplet and observation for subsequent analyses (reasoning explained in Methods). Comparing the controls with the shaded triplefins showed that the stimulus effect depended on the hat treatment in the lab and in the north-facing field tanks, but not in the south-facing field tanks (LMM: *hat treatment×stimulus* $p_{lab}$ = 0.017, $p_{field-north}$ = 0.038, $p_{field-south}$ = 0.248, electronic supplementary material, tables S1b–S2b-c). The significant interaction terms resulted from shaded individuals staying significantly closer to the scorpionfish than the controls (LMM scorpionfish: *hat treatment* $p_{lab}$ < 0.0001, $p_{field-north}$ = 0.011, electronic supplementary material, tables S1c-S2b), while this was not the case for the stone (LMM stone: *hat treatment* $p_{lab}$ = 0.21, $p_{field-north}$ = 0.097, details not shown).

### (b) Immediate response of hatted individuals to a scorpionfish (second field experiment)

Following release in the tank, many triplefins started hopping towards the scorpionfish, presumably attracted by the shaded display compartment. The first distance measurement, taken approximately 1 min after release (figure 3), showed that clear-hatted individuals were more hesitant to come close than shaded triplefins. Only nine (21%) clear-hatted versus eighteen (47%) shaded individuals had come closer than approximately 7 cm (Fisher's exact test: odds ratio = 0.31, 95% CI = 0.10–0.88, $p$ = 0.019). This is the distance at which

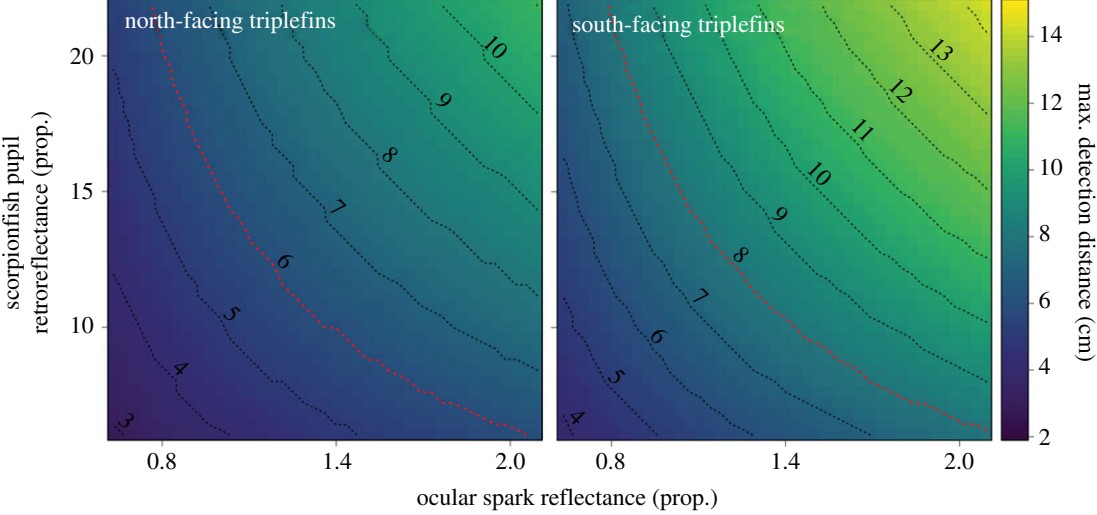

**Figure 4.** Visual modelling summary showing detection distances (colour) by a triplefin of reflections in a scorpionfish pupil induced by a blue ocular spark. The outcome is shown as a function of spark reflectance and scorpionfish pupil retroreflectance, and separated for north- and south-facing orientations. Note that reflectance is expressed as proportion relative to a diffuse white standard, not as %: values above 1 indicate a reflectance of greater than 100%. The dashed lines at 6 cm (left panel) and 8 cm (right panel) represent the average detection distances that were combined in the 7 cm value shown in figure 3. Values were obtained from calculating the achromatic Michelson contrast based on the sum of triplefin cone-catches of the double cones between 1 and 15 cm, and identifying the distance at which the contrast reached the achromatic contrast threshold of *T. delaisi* [23] (see Material and methods). (Online version in colour.)

active photolocation with unobstructed ocular sparks is predicted to function under average conditions (next section). Following the initial approach, individuals gradually retreated to the opposite half of the tank, but clear-hatted individuals did so approximately 20 min earlier than shading-hatted fish (GLMM: *hat treatment p = 0.034, time p < 0.0001, hat treatment × time p = 0.036*) (electronic supplementary material, table S3). Both treatments reached a similar distance after approximately 50 min. In contrast to the first field experiment, the random predictor *orientation* had no effect, perhaps a consequence of the lack of a long acclimation time, the shorter distance available to move away (50 cm versus 125 cm earlier), and the shorter observation time window.

### (c) Visual modelling

While ocular sparks did not generate chromatic contrast above the discriminability threshold at any distance between the triplefin and the scorpionfish, achromatic Michelson contrasts exceeded the detection thresholds across a broad range of conditions (figure 4). For comparison, identical calculations for spark-generated contrast changes in a scorpionfish's iris rather than its pupil showed no perceptible effect under any of the tested conditions (not shown). This confirms that subocular light emission is subtle and can only generate detectable contrasts in strong reflectors, e.g. retroreflective eyes. For north-facing triplefins, the reflections induced by an ocular spark in a scorpionfish's pupil would be detectable up to 6 cm under average conditions, and up to 10 cm for higher values of ocular spark radiance and scorpionfish pupil retroreflectance. These distances increased by 2–3 cm for south-facing triplefins. The average detection range of 6–8 cm coincides well with the initial approach distances in the third experiment (figure 3). Note that these are distances to the eye of a scorpionfish. A triplefin that approaches a scorpionfish from the side or back can be only a few centimetres from the predator's eye and still be outside the striking distance of its mouth.

## 4. Discussion

Our results provide a first proof of concept for diurnal active photolocation of a cryptobenthic predator. The suppression of ocular sparks significantly reduced the distance triplefins kept from a well-camouflaged scorpionfish in three experiments. The effect was strongest in the laboratory experiment, where the behaviour was assessed after 14 h acclimatisation. A replicate experiment under realistic conditions in the field showed more variation due to confounding factors, but the same effect was still present in one orientation (north), but was not in the other orientation (south), where all treatments responded strongly to the scorpionfish. This suggests that predator visual detection was easy under these conditions, and independent of the hatting treatment. A second field experiment, where fish were given no acclimation time, assessed the exploratory response of individual fish to a scorpionfish immediately after they were released into the tank, again revealing a significant difference in the predicted direction. These experiments, however, did not specifically test whether the scorpionfish's pupil was the target of detection. We, therefore, used visual modelling and showed that an ocular spark is strong enough to increase the achromatic contrast in a scorpionfish's pupil above a triplefins's perception threshold over distances that coincide well with the nearest approaches seen in the third field experiment.

As has been made clear before [15], this study once more indicates that diurnal active photolocation is not fail-proof, as shown by the predictions from the visual modelling and by the fact that even clear-hatted triplefins approached scorpionfish within ranges that may fall within the striking range of a scorpionfish [41–43]. Diurnal active photolocation is also not strictly required for detecting a scorpionfish, as shown by the response of shaded triplefins to scorpionfish relative to stones. Since communication between individuals was possible in the first two experiments, shaded fish may have been influenced by the behaviour of their control-treated

conspecifics. Yet, the third experiment that tested only single individuals showed the same effects. We conclude that diurnal active photolocation is a mechanism that supplements a triplefin's ability to detect scorpionfish.

The chromatophore patch on the lower iris of the triplefin *T. delaisi* on which the ocular spark is focused behaves like a diffuse, Lambertian reflector in the equatorial plane of vision [15]. This produces a light field that covers most of the hemispherical zone over a short distance, as seen by a single eye. In lantern and flashlight fish, subocular light organs are also considered diffuse sources [8,44]. However, many other fish possess silvery irides with near-specular properties. Such reflectors are more directional, presumably allowing specific illumination of objects over greater distances. Yet, this property also may attract the attention of visual predators. Trade-offs like these may explain variation seen among fish in types of ocular light redirection [13,45]. As for the to-be-detected target, highly reflective structures such as retroreflective eyes in predatory fish [12,18] or reflective eyecups or ommatidia in crustacean prey [13,15] are also common and diverse. Since none of these have been studied in this context, it is too early to speculate which combination of observer/target reflectors and ambient conditions is likely to function. Yet, it is clear that the building blocks required for diurnal active photolocation are widespread. Many species may be using it to detect their predators.

Most marine cryptobenthic predators show eye adaptations that modify the visual appearance of their pupils. Stonefish (*Synanceia*) and frogfish (*Antennarius*) have small pupils for their body size [46]. Other species have skin flaps that cover the pupil as in shovel-nosed rays (*Aptchotrema rostrate*) [1], flatheads (Platycephalidae) and stargazers (Uranoscopidae) or possess slit-like pupils as in some flatheads (*Thysanophrys*, Platycephalidae), flounders (*Bothus*) and sandperches (*Parapercis*) [46]. In lionfishes (*Pterois*, Scorpaenidae) the eyes are embedded in a black vertical band [46]. Other scorpionfish (Scorpaenidae) and devil stingers (*Inimicus*, Synanceidae) conceal their black pupils by a combination of different types of eyeshine [18]. All of these traits reduce pupil size, distort its shape, or mask its presence. Since eyes are commonly used for face detection [47–50], modifications like these are likely to hamper pupil detection [51]. At least in species featuring diurnal retroreflective eyeshine (e.g. Scorpaenidae), the above-mentioned eye adaptations of cryptobenthic predators may also represent counteradaptations to observers that use diurnal active photolocation.

Our work illustrates that the visual interactions between cryptobenthic predators and their prey remain poorly understood, presumably because direct interactions are rarely observed in the field and take place at small spatial scales unfamiliar and counterintuitive to humans [52,53].

Data accessibility. Data are available from the Dryad Digital Repository: https://doi.org/10.5061/dryad.3bk3j9kdn [54].

Authors' contributions. N.K.M., R.F., M.S., P.-P.B. and U.K.H. conceived the study. R.F. developed the hatting technique. N.K.M., J.D., M.S., U.K.H., R.F. and P.-P.B. conceptualized the experiments. J.D. collected the laboratory data. M.S. and U.K.H. collected the field data, with assistance from the whole crew. M.S. and N.A. analysed the experimental data. P.-P.B. and M.S. developed and ran the visual model using spectroradiometric data collected by M.S. and U.K.H. The manuscript was written by N.K.M., M.S., P.-P.B., R.F. and J.D. All authors edited and approved the manuscript.

Competing interests. We declare we have no competing interests.

Funding. N.K.M. was supported by the Deutsche Forschungsgemeinschaft (Koselleck Grant Mi 482/13-1) and the Volkswagen Foundation (Experiment! grant no. Az. 89148 and Az. 91816). P.-P.B. was funded by the Natural Sciences and Engineering Research Council of Canada (Postdoctoral Fellowship, grant no. 471704 - 2015).

Acknowledgements. We are indebted to the attendants of a workshop in November 2014 in Tübingen with Connor M. Champ, João Coimbra, Colin B. Jack, Sönke Johnsen, Almut Kelber, Melissa G. Meadows, Daniel Osorio, Shelby Temple and Annette Werner. Thanks to Martin J. How and Jennifer R. Hodge for useful suggestions on an earlier draft. Jonas Dornbach, Thomas Griessler, Katharina Hiemer, Michael Karcz, Valentina Richter, Peter Tung, Sabine Urban, Laura Warmuth and Florian Wehrberger supported data collection in the field. Gregor Schulte provided creative and technical support. Thanks to Pierre Lejeune, director of STARESO, and his staff, for providing excellent working conditions.

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
