## [Reviewer comments · Proceedings of the Royal Society B: Biological Sciences]

Review History

RSPB-2019-0473.R0 (Original submission)

Review form: Reviewer 1 (Jolyon Troscianko)

Recommendation

Accept with minor revision (please list in comments)

Scientific importance: Is the manuscript an original and important contribution to its field?

Excellent

General interest: Is the paper of sufficient general interest?

Excellent

Quality of the paper: Is the overall quality of the paper suitable?

Excellent

Is the length of the paper justified?

Yes

Should the paper be seen by a specialist statistical reviewer?

No

Do you have any concerns about statistical analyses in this paper? If so, please specify them explicitly in your report.

No

It is a condition of publication that authors make their supporting data, code and materials available - either as supplementary material or hosted in an external repository. Please rate, if applicable, the supporting data on the following criteria.

Is it accessible?

Yes

Is it clear?

No

Is it adequate?

Yes

Do you have any ethical concerns with this paper?

No

Comments to the Author

Many animals are easy to spot at night by using a torch and looking for their eye-shine. The idea that fish could be doing the same thing to detect predators and prey is a fascinating prospect. This manuscript presents a large body of work to demonstrate how tripplefin fish could use special reflective irises to create "flashes" or "sparks" required to detect the eye-shine of nearby predators.

I know that this issue is highly controversial in some circles, which is why I'm glad to see the authors have been so thorough in their analysis, and I believe it is an exciting new mode of predator-prey interaction worthy of publication in Proceedings B.

There is no reason in principle why the underlying physics cannot produce eye-shine (at least at some distance), particularly given the incredible range of photonic structures seen in nature. Fish have large eyes which protrude from the sides of their heads. Lenses such as this will focus substantial down-welling light outside of the retina, and onto the iris instead (so there is the collection of lots of light). Then a reflective surface/mirror on the iris can be used to reflect that light. Adding motor control to the mirror will mean it can be pointed in different directions. The authors have performed all of the modelling required to show that this mechanism can work in principle, and have also performed a range of behavioural tests to show that the tripplefin can use the mechanism to avoid predatory fish.

Overall I have no major concerns about the methods or analysis. I have skimmed through the R code to try to work out how the modelling works, and it seems sensible. The stats seem good (though see minor points below).

General Comments:

An alternative hypothesis (which the authors don't mention) could be that the tripplefin use their sparks to signal to the predators that they can see them (predator signalling and inspection is quite common in fish). Given the tripplefin of all treatments could avoid the predators, what

would the predictions of this theory be, and can it be tested with the existing data? This might also lead to expected differences between the fish of different treatments, though it's not immediately clear to me what those predictions would be, and in any case the ability to avoid the presence of a predator is a more parsimonious explanation for the author's results.

Regarding the ability of tripplefish of all treatment types to detect scorpionfish, presumably the scorpionfish moved every now & then, so this would be a clear cue for the tripplefish that they are not a rock?

Methods: The distance measurement method used is certainly not optimal (i.e. judging distance based on a ruler, which will have a different angle to that being measured). Ideally the researchers would have used cameras to record the positions of the fish at regular intervals, and then distances could be measured from the images (near infrared images would also make it impossible to distinguish between hat treatments). The repeatability analysis (comparing estimates from the two divers) goes some way to addressing this, however we must assume both divers were not naive to the hypothesis being tested. So if there were any measurement bias, all we know is that both divers were equally bias (which is highly unlikely, but possible).

How was the retro-reflectance of the scorpionfish eye measured, from how many samples? Wouldn't you need an ophthalmoscope-based setup in a darkroom to measure this? The reflectance used in the code seems to just be the first column in the file (about 5% reflectance at 500nm, which sounds plausible, but isn't labelled). This value is quite critical to the calculations though, so needs much more clarity. I think figure 4 is really valuable here to highlight the parameter-space. e.g. even if the retroreflectance from the scorpionfish pupil is very low, and the spark is within the range measured, the retroreflection should still be visible 5-6cm away. I think the authors should highlight this graph earlier on (with the equations), and maybe specify what the measured parameter ranges were.

Specific comments:

The grammar needs sorting out in a few places (particularly the intro).

L215 Was a different distribution used because the tank was smaller (making the distance data more bounded and less normal?) edit: you explain this later, maybe say "see below", or just don't mention it here.

L209 If you don't average the control-hat and no-hat fish do you get the same results? Or is the sample size too small? It would at least be good to add this full analysis if possible (e.g. as supplementary data), which should at least show the same trends, even if the sample size is small.

I would like more clear drawings/photos/descriptions of the field tanks. I'm slightly confused as to how exactly the control stone and scorpion fish are presented. Edit: I see in the SEM there are nice figures, maybe point towards these in the text.

L228-230 I'm confused as to why models for north and south oriented tanks were calculated separately given orientation (and its interactions) was added to the model above (L225). Do you mean you re-ran the same single model but specified different contrasts within it, or did you split the data and run two models? Edit: this makes a bit more sense after seeing the different effects in figure 4.

L280 what is a liquid light guide?

L305 Equations 2 & 6 are in the SEM.

Your equations don't make it clear that the calculation is repeated at each 1nm increment based on reflectance spectra before summing to cone-catch quanta (e.g. maybe the equations should have the relevant functions added).

L414. Maybe say "face detection" instead? Face recognition implies identification of an individual based on their face (at least in other spheres of science). Fish can do thi, but I think that's not the point the authors mean.

Jolyon Troscianko

Review form: Reviewer 2

Recommendation

Reject - article is scientifically unsound

Scientific importance: Is the manuscript an original and important contribution to its field?

Marginal

General interest: Is the paper of sufficient general interest?

Acceptable

Quality of the paper: Is the overall quality of the paper suitable?

Marginal

Is the length of the paper justified?

Yes

Should the paper be seen by a specialist statistical reviewer?

Yes

Do you have any concerns about statistical analyses in this paper? If so, please specify them explicitly in your report.

Yes

It is a condition of publication that authors make their supporting data, code and materials available - either as supplementary material or hosted in an external repository. Please rate, if applicable, the supporting data on the following criteria.

Is it accessible?

Yes

Is it clear?

No

Is it adequate?

Yes

Do you have any ethical concerns with this paper?

No

Comments to the Author

In this study, the authors test the effect of an ocular reflector on the predator detection abilities of the triplefin in the lab and in the field using a predator (scorpionfish) or a predator mimic (rock). I have a number of major concerns regarding the concepts explored in this paper as well as the methodologies used to test these concepts.

The authors use the concept and term “active sensing” incorrectly throughout this manuscript. Active sensing is the use of self-generated energy to sense the environment (Nelson and MacIver, 2006). Examples of active sensing that the authors present are the light organs of lantern fish and flashlight fish where these animals produce the luminescence by which they hunt. These examples are not comparable to the light reflected from the iris in triplefins. Light reflected from the iris is passive, even if the animals can sense it with photoreceptors in or near their eyes.

The no-hat and clear-hat controls should not be treated as the same group. They are different types of controls.

Another major concern is the use of the concept “diurnal active photolocation”. The authors treat “regular vision” as a static and rigid sense instead of the dynamic ability to detect light under many environmental conditions and integrate that information with other cues (such as predator chemical cues). The ability to detect a predator visually is part of vision, and the use of vision to interact with the environment is a whole field in and of itself (visual ecology). Many visual systems are adapted to such processes. Vision, plus the ability to detect the chemical cues of the predator, make this a system where the animals can rely on the integration of their sensory systems to evade a predator.

Perhaps changing the light that is entering the eye by placing either the shaded or unshaded hat changes the visual environment simply by reducing the amount of light that reaches the eye. Fish eyes are designed to reflect light so that superfluous sunlight cannot reach the photoreceptors in order to reduce the signal-to-noise ratio (Land and Nilsson, 2012). Thus, by changing the amount of light entering the eye, this changes the perception of the environment in ways that may alter resolution and/or sensitivity. This may result in poorer vision and the lack of ability to properly detect a predator visually, causing the prey to move closer to detect the predator by other modes of sensation such as chemosensation.

The authors fail to mention that triplefins have at least four chromatophore patches on their iris. They refer only to the most ventral patch. Is only the patch below the lens useful? What about the breeding male phenotype that has no patch? Are they less able to detect predators? This would be interesting to test given the polymorphic nature of these males.

Why do animals need to acclimate to visual stimuli for 14+ hours? I suspect that the long acclimation period allowed animals to acclimate to the chemical cue of the predator, rather than acclimate to any visual stimulus. Animals should acclimate to visual stimuli quickly, if this is the only sensory cue that they are using.

I also find the timing of data collection for the lab experiments to be odd. There is so much unaccounted time that the animals are not being observed where they may be moving around, exploring, or interacting. What is the significance of recording their positions for 5 minutes every 2-3 hours? It is quite understandable to have field experiments be timed as they are since bottom-time is restrictive for data collection in this experiment.

Review form: Reviewer 3

Recommendation

Reject – article is scientifically unsound

Scientific importance: Is the manuscript an original and important contribution to its field?

Good

General interest: Is the paper of sufficient general interest?

Acceptable

Quality of the paper: Is the overall quality of the paper suitable?

Marginal

Is the length of the paper justified?

Yes

Should the paper be seen by a specialist statistical reviewer?

Yes

Do you have any concerns about statistical analyses in this paper? If so, please specify them explicitly in your report.

Yes

It is a condition of publication that authors make their supporting data, code and materials available - either as supplementary material or hosted in an external repository. Please rate, if applicable, the supporting data on the following criteria.

Is it accessible?

Yes

Is it clear?

Yes

Is it adequate?

Yes

Do you have any ethical concerns with this paper?

No

Comments to the Author

This paper examines the exciting hypothesis that some fish may use an ocular spark to improve their detection of predators. Using triplefin fish as an example, the authors use lab, field, and visual modelling methods to investigate this hypothesis. By making triplefins unable to produce optic sparks through shaded hats, the authors examine if hatted fish show a decreased ability to respond to a camouflaged predator (via smaller approach distances). Although I find the idea and potential mechanism exciting, in my opinion both the visual modelling and behavioral data have issues (some with interpretation), and as a result do not fully support the claims made in this paper.

Although the paper presents the behavioral data first, I find it easier to start with the visual modelling. The visual modelling section found that “ocular sparks did not generate chromatic contrast above discriminability threshold” and that achromatic contrasts were only exceeded at values smaller than 6-8cms depending on the location. In many ways, these values represent best

case scenarios. For example, 1) the Michelson contrast threshold of 0.008 used in the modelling is the most sensitive found in a previous study and may be less sensitive depending of the angle of the stimulus [cited ref 13], 2) the radiance of the ocular spark used was “the highest value for each fish relative to the standard”, and 3) the ocular spark was only measured at 45° despite possibly (likely?) having angular dependence. Unfortunately, the authors do not report what % of photons are redirected via the optical spark, nor what % of photons that hit a scorpionfish pupil and iris is actually reflected. I tried to find this information in the supplementary tables, but am uncertain if I am interpreting it correctly because the columns of ESM13 and ESM15 are not labeled (I’m assuming its for different individuals, but may be wrong). Reporting these values would go a long way in understanding just how subtle this light is in a natural condition. That chromatic contrast was not changed (despite the fascinating pictures in Figure 1 d-f) and that the achromatic contrast threshold is only reached at very short distances suggests that this light production may be too subtle to be functional.

With the visual modelling suggesting a limited – although feasible --- range of function, one would like strong behavioral data to support it. The modelling suggests that if there is any predator avoidance effect from the optic spark that it should occur only at very short distances. As such I’d love to minimum approach distance to the predator, but this data was not collected, and just average distance is shown instead. This makes it hard to tell if the difference between treatments (when present) is occurring at a similar distance to what is predicted via visual modeling. Video analysis of the initial approach to the predator would be much more convincing than locations at 5 spaced times over the course of a day. Without this data I feel uncomfortable attributing the differences to the optic spark as there could be various other factors involved (including those suggested by the authors such as the range reduction from wearing a hat, or the impossibility to blind the experiment). There may be some way to pull out this data from Figure 3 (or at least show immediate minimum distance) but as currently presented it is not there.

Additionally, I have a few statistical questions (I do have experience with these tests, but am not an expert). I’m used to papers that use GLMM or LMM use model fitting to pull out the most parsimonious models, but this has not been done (or at least is not present in the paper/supplementary materials or I missed it). I’m also uncomfortable that the authors “averaged the data of the two control-treated fish per triplet” because there was no significant difference between them. Just because a difference is not significant doesn’t mean that there isn’t an effect. Considering all the other factors influenced by wearing a hat, I’d love to see the data presented in a clear fashion where I can compare between the transparent and opaque hats, but this is not possible. Lastly, I find it weird that the authors present all their data as interactions (Treatment x Stimulus). Because the stone data did not vary significantly in any treatment it seems appropriate to present just the scorpionfish data without getting into the more complicated interactions. This is especially relevant for Figure 2b, where it appears the statistically significant difference in the Stimulus x Treatment interaction is driven in part by a slight (likely random) preference for the shading hat fish to be farther away from a stone. The scorpionfish data does not look significant by itself.

Considering these factors all together, in my opinion the data presented here does not do enough to back the claims of the paper. Figure 2C gives a negative result, Figure 2B appears to be driven to significance by stone differences (also likely some outlier fish never approaching?), and none of the figures present information about the distance at which this behavioral choice is made.

Overall this idea is promising and biologically exciting. However I’d like to see 1) reporting of the % of photons produced via the optical spark and the % reflected by the scorpionfish pupil, 2) having behavioral trials that show the distance of predator avoidance roughly aligning with the visual modelling, and 3) stronger overall behavioral data.

Minor Comments:

Introduction: Paragraphs! I feel the introduction also needs a lot more context. Lines 38-56 are the only ones currently providing this, the rest just describe the experiment. I’d also suggest defining more words to reach a broader audience (retroreflective, coaxial, etc.)

Ln 41-43 A figure showing the proposed mechanism would help your reader a lot here.

Discussion

Ln 385 Proof of principle seems to strong here

Ln 390-392 If the mechanism only works when already in range of the scorpionfish, is it really biologically relevant?

Ln 393 I'd love to see more discussion of the limitations to triplfins before going into other species later in this paragraph. What are the limitations? The angular distribution of light? Depths where it would be effective? Etc.

Review form: Reviewer 4

Recommendation

Accept as is

Scientific importance: Is the manuscript an original and important contribution to its field?

Excellent

General interest: Is the paper of sufficient general interest?

Good

Quality of the paper: Is the overall quality of the paper suitable?

Excellent

Is the length of the paper justified?

Yes

Should the paper be seen by a specialist statistical reviewer?

No

Do you have any concerns about statistical analyses in this paper? If so, please specify them explicitly in your report.

No

It is a condition of publication that authors make their supporting data, code and materials available - either as supplementary material or hosted in an external repository. Please rate, if applicable, the supporting data on the following criteria.

Is it accessible?

Yes

Is it clear?

Yes

Is it adequate?

Yes

Do you have any ethical concerns with this paper?

No

Comments to the Author

Comments:

This manuscript describes interesting new data on the functionality of diurnal active photolocation. The authors had previously shown that the benthic triplefin, *Tripterygion delaisi*, controlled redirection of downwelling sunlight using their iris, generating a phenomenon called "ocular spark". In this work, they hypothesize that ocular sparks improve visual detection of nearby cryptic organisms, a process they called "diurnal active photolocation". To test this hypothesis, these authors experimentally suppressed light redirection using the benthic triplefin *Tripterygion delaisi* as a model. They manipulated light redirection through an unusual approach, using shaded hats glued on the fish heads. The authors presented results from the effect of hat treatment on the average distance from the stimulus (stone or scorpionfish), immediate responses from the predator scorpionfish, and color difference metrics/chromatic contrasts are measured as just-noticeable differences (JNDs).

On the first experiment, the authors performed a very careful and controlled experiment, to set the stage for the second (field) experiment, a more robust approach, using more specimens for the analysis. Both setting seemed to be well planned and well written. The results of these experiments led the authors to validate the results implementing visual models to compute the contrast change in the pupil of a scorpionfish as perceived by an untreated triplefin when producing an ocular spark.

Although the data appear sound, I have a minor concern concerning the interpretation of the results based on the recorded movements, not taking into account any behavior analysis; however, the results support all the conclusions and their hypothesis of ocular spark improving visual detection of cryptic organisms.

Decision letter (RSPB-2019-0473.R0)

04-Jun-2019

Dear Professor Michiels:

I am writing to inform you that your manuscript RSPB-2019-0473 entitled "Active sensing with light improves predator detection in a diurnal fish" has, in its current form, been rejected for publication in Proceedings B.

This action has been taken on the advice of referees, who have recommended that substantial revisions are necessary. With this in mind we would be happy to consider a resubmission, provided the comments of the referees are fully addressed. However please note that this is not a provisional acceptance.

Sincerely,
 Dr Daniel Costa
 mailto: proceedingsb@royalsociety.org

Associate Editor
 Board Member: 1

Comments to Author:

We have now heard from four reviewers -- and it is clear that there is a split decision with respect to your manuscript. I am recommending rejection but I hope that you will consider re-submitting a revised manuscript that deals with the critical comments of the reviewers.

Reviewer(s)' Comments to Author:

Referee: 1

Comments to the Author(s)

Many animals are easy to spot at night by using a torch and looking for their eye-shine. The idea that fish could be doing the same thing to detect predators and prey is a fascinating prospect. This manuscript presents a large body of work to demonstrate how tripplefin fish could use special reflective irises to create "flashes" or "sparks" required to detect the eye-shine of nearby predators.

I know that this issue is highly controversial in some circles, which is why I'm glad to see the authors have been so thorough in their analysis, and I believe it is an exciting new mode of predator-prey interaction worthy of publication in Proceedings B.

There is no reason in principle why the underlying physics cannot produce eye-shine (at least at some distance), particularly given the incredible range of photonic structures seen in nature. Fish have large eyes which protrude from the sides of their heads. Lenses such as this will focus substantial down-welling light outside of the retina, and onto the iris instead (so there is the collection of lots of light). Then a reflective surface/mirror on the iris can be used to reflect that light. Adding motor control to the mirror will mean it can be pointed in different directions. The authors have performed all of the modelling required to show that this mechanism can work in principle, and have also performed a range of behavioural tests to show that the tripplefin can use the mechanism to avoid predatory fish.

Overall I have no major concerns about the methods or analysis. I have skimmed through the R code to try to work out how the modelling works, and it seems sensible. The stats seem good (though see minor points below).

General Comments:

An alternative hypothesis (which the authors don't mention) could be that the tripplefins use their sparks to signal to the predators that they can see them (predator signalling and inspection is quite common in fish). Given the tripplefins of all treatments could avoid the predators, what would the predictions of this theory be, and can it be tested with the existing data? This might also lead to expected differences between the fish of different treatments, though it's not immediately clear to me what those predictions would be, and in any case the ability to avoid the presence of a predator is a more parsimonious explanation for the author's results.

Regarding the ability of tripplefins of all treatment types to detect scorpionfish, presumably the scorpionfish moved every now & then, so this would be a clear cue for the tripplefins that they are not a rock?

Methods: The distance measurement method used is certainly not optimal (i.e. judging distance based on a ruler, which will have a different angle to that being measured). Ideally the researchers would have used cameras to record the positions of the fish at regular intervals, and then distances could be measured from the images (near infrared images would also make it impossible to distinguish between hat treatments). The repeatability analysis (comparing estimates from the two divers) goes some way to addressing this, however we must assume both divers were not naive to the hypothesis being tested. So if there were any measurement bias, all we know is that both divers were equally bias (which is highly unlikely, but possible).

How was the retro-reflectance of the scorpionfish eye measured, from how many samples? Wouldn't you need an ophthalmoscope-based setup in a darkroom to measure this? The reflectance used in the code seems to just be the first column in the file (about 5% reflectance at 500nm, which sounds plausible, but isn't labelled). This value is quite critical to the calculations though, so needs much more clarity. I think figure 4 is really valuable here to highlight the parameter-space. e.g. even if the retroreflectance from the scorpionfish pupil is very low, and the spark is within the range measured, the retroreflection should still be visible 5-6cm away. I think the authors should highlight this graph earlier on (with the equations), and maybe specify what the measured parameter ranges were.

Specific comments:

The grammar needs sorting out in a few places (particularly the intro).

L215 Was a different distribution used because the tank was smaller (making the distance data more bounded and less normal?) edit: you explain this later, maybe say "see below", or just don't mention it here.

L209 If you don't average the control-hat and no-hat fish do you get the same results? Or is the sample size too small? It would at least be good to add this full analysis if possible (e.g. as supplementary data), which should at least show the same trends, even if the sample size is small.

I would like more clear drawings/photos/descriptions of the field tanks. I'm slightly confused as to how exactly the control stone and scorpion fish are presented. Edit: I see in the SEM there are nice figures, maybe point towards these in the text.

L228-230 I'm confused as to why models for north and south oriented tanks were calculated separately given orientation (and its interactions) was added to the model above (L225). Do you mean you re-ran the same single model but specified different contrasts within it, or did you split

the data and run two models? Edit: this makes a bit more sense after seeing the different effects in figure 4.

L280 what is a liquid light guide?

L305 Equations 2 & 6 are in the SEM.

Your equations don't make it clear that the calculation is repeated at each 1nm increment based on reflectance spectra before summing to cone-catch quanta (e.g. maybe the equations should have the relevant functions added).

L414. Maybe say "face detection" instead? Face recognition implies identification of an individual based on their face (at least in other spheres of science). Fish can do thi, but I think that's not the point the authors mean.

Jolyon Troscianko

Referee: 2

Comments to the Author(s)

In this study, the authors test the effect of an ocular reflector on the predator detection abilities of the triplefin in the lab and in the field using a predator (scorpionfish) or a predator mimic (rock). I have a number of major concerns regarding the concepts explored in this paper as well as the methodologies used to test these concepts.

The authors use the concept and term "active sensing" incorrectly throughout this manuscript. Active sensing is the use of self-generated energy to sense the environment (Nelson and MacIver, 2006). Examples of active sensing that the authors present are the light organs of lantern fish and flashlight fish where these animals produce the luminescence by which they hunt. These examples are not comparable to the light reflected from the iris in triplefins. Light reflected from the iris is passive, even if the animals can sense it with photoreceptors in or near their eyes.

The no-hat and clear-hat controls should not be treated as the same group. They are different types of controls.

Another major concern is the use of the concept "diurnal active photolocation". The authors treat "regular vision" as a static and rigid sense instead of the dynamic ability to detect light under many environmental conditions and integrate that information with other cues (such as predator chemical cues). The ability to detect a predator visually is part of vision, and the use of vision to interact with the environment is a whole field in and of itself (visual ecology). Many visual systems are adapted to such processes. Vision, plus the ability to detect the chemical cues of the predator, make this a system where the animals can rely on the integration of their sensory systems to evade a predator.

Perhaps changing the light that is entering the eye by placing either the shaded or unshaded hat changes the visual environment simply by reducing the amount of light that reaches the eye. Fish eyes are designed to reflect light so that superfluous sunlight cannot reach the photoreceptors in order to reduce the signal-to-noise ratio (Land and Nilsson, 2012). Thus, by changing the amount of light entering the eye, this changes the perception of the environment in ways that may alter resolution and/or sensitivity. This may result in poorer vision and the lack of ability to properly detect a predator visually, causing the prey to move closer to detect the predator by other modes of sensation such as chemosensation.

The authors fail to mention that triplefins have at least four chromatophore patches on their iris. They refer only to the most ventral patch. Is only the patch below the lens useful? What about the breeding male phenotype that has no patch? Are they less able to detect predators? This would be interesting to test given the polymorphic nature of these males.

Why do animals need to acclimate to visual stimuli for 14+ hours? I suspect that the long acclimation period allowed animals to acclimate to the chemical cue of the predator, rather than acclimate to any visual stimulus. Animals should acclimate to visual stimuli quickly, if this is the only sensory cue that they are using.

I also find the timing of data collection for the lab experiments to be odd. There is so much unaccounted time that the animals are not being observed where they may be moving around, exploring, or interacting. What is the significance of recording their positions for 5 minutes every 2-3 hours? It is quite understandable to have field experiments be timed as they are since bottom-time is restrictive for data collection in this experiment.

Referee: 3

Comments to the Author(s)

This paper examines the exciting hypothesis that some fish may use an ocular spark to improve their detection of predators. Using triplefin fish as an example, the authors use lab, field, and visual modelling methods to investigate this hypothesis. By making triplefins unable to produce optic sparks through shaded hats, the authors examine if hatted fish show a decreased ability to respond to a camouflaged predator (via smaller approach distances). Although I find the idea and potential mechanism exciting, in my opinion both the visual modelling and behavioral data have issues (some with interpretation), and as a result do not fully support the claims made in this paper.

Although the paper presents the behavioral data first, I find it easier to start with the visual modelling. The visual modelling section found that “ocular sparks did not generate chromatic contrast above discriminability threshold” and that achromatic contrasts were only exceeded at values smaller than 6-8cms depending on the location. In many ways, these values represent best case scenarios. For example, 1) the Michelson contrast threshold of 0.008 used in the modelling is the most sensitive found in a previous study and may be less sensitive depending of the angle of the stimulus [cited ref 13], 2) the radiance of the ocular spark used was “the highest value for each fish relative to the standard”, and 3) the ocular spark was only measured at 45° despite possibly (likely?) having angular dependence. Unfortunately, the authors do not report what % of photons are redirected via the optical spark, nor what % of photons that hit a scorpionfish pupil and iris is actually reflected. I tried to find this information in the supplementary tables, but am uncertain if I am interpreting it correctly because the columns of ESM13 and ESM15 are not labeled (I’m assuming its for different individuals, but may be wrong). Reporting these values would go a long way in understanding just how subtle this light is in a natural condition. That chromatic contrast was not changed (despite the fascinating pictures in Figure 1 d-f) and that the achromatic contrast threshold is only reached at very short distances suggests that this light production may be too subtle to be functional.

With the visual modelling suggesting a limited – although feasible --- range of function, one would like strong behavioral data to support it. The modelling suggests that if there is any predator avoidance effect from the optic spark that it should occur only at very short distances. As such I’d love to minimum approach distance to the predator, but this data was not collected, and just average distance is shown instead. This makes it hard to tell if the difference between treatments (when present) is occurring at a similar distance to what is predicted via visual modeling. Video analysis of the initial approach to the predator would be much more convincing

than locations at 5 spaced times over the course of a day. Without this data I feel uncomfortable attributing the differences to the optic spark as there could be various other factors involved (including those suggested by the authors such as the range reduction from wearing a hat, or the impossibility to blind the experiment). There may be some way to pull out this data from Figure 3 (or at least show immediate minimum distance) but as currently presented it is not there.

Additionally, I have a few statistical questions (I do have experience with these tests, but am not an expert). I'm used to papers that use GLMM or LMM use model fitting to pull out the most parsimonious models, but this has not been done (or at least is not present in the paper/supplementary materials or I missed it). I'm also uncomfortable that the authors "averaged the data of the two control-treated fish per triplet" because there was no significant difference between them. Just because a difference is not significant doesn't mean that there isn't an effect. Considering all the other factors influenced by wearing a hat, I'd love to see the data presented in a clear fashion where I can compare between the transparent and opaque hats, but this is not possible. Lastly, I find it weird that the authors present all their data as interactions (Treatment x Stimulus). Because the stone data did not vary significantly in any treatment it seems appropriate to present just the scorpionfish data without getting into the more complicated interactions. This is especially relevant for Figure 2b, where it appears the statistically significant difference in the Stimulus x Treatment interaction is driven in part by a slight (likely random) preference for the shading hat fish to be farther away from a stone. The scorpionfish data does not look significant by itself.

Considering these factors all together, in my opinion the data presented here does not do enough to back the claims of the paper. Figure 2C gives a negative result, Figure 2B appears to be driven to significance by stone differences (also likely some outlier fish never approaching?), and none of the figures present information about the distance at which this behavioral choice is made.

Overall this idea is promising and biologically exciting. However I'd like to see 1) reporting of the % of photons produced via the optical spark and the % reflected by the scorpionfish pupil, 2) having behavioral trials that show the distance of predator avoidance roughly aligning with the visual modelling, and 3) stronger overall behavioral data.

Minor Comments:

Introduction: Paragraphs! I feel the introduction also needs a lot more context. Lines 38-56 are the only ones currently providing this, the rest just describe the experiment. I'd also suggest defining more words to reach a broader audience (retroreflective, coaxial, etc.)

Ln 41-43 A figure showing the proposed mechanism would help your reader a lot here.

Discussion

Ln 385 Proof of principle seems to strong here

Ln 390-392 If the mechanism only works when already in range of the scorpionfish, is it really biologically relevant?

Ln 393 I'd love to see more discussion of the limitations to triplfins before going into other species later in this paragraph. What are the limitations? The angular distribution of light? Depths where it would be effective? Etc.

Referee: 4

Comments to the Author(s)

Comments:

This manuscript describes interesting new data on the functionality of diurnal active photolocation. The authors had previously shown that the benthic triplefin, *Tripterygion delaisi*, controlled redirection of downwelling sunlight using their iris, generating a phenomenon called "ocular spark". In this work, they hypothesize that ocular sparks improve visual detection of nearby cryptic organisms, a process they called "diurnal active photolocation". To test this hypothesis, these authors experimentally suppressed light redirection using the benthic triplefin

Tripterygion delaisi as a model. They manipulated light redirection through an unusual approach, using shaded hats glued on the fish heads. The authors presented results from the effect of hat treatment on the average distance from the stimulus (stone or scorpionfish), immediate responses from the predator scorpionfish, and color difference metrics/chromatic contrasts are measured as just-noticeable differences (JNDs).

On the first experiment, the authors performed a very careful and controlled experiment, to set the stage for the second (field) experiment, a more robust approach, using more specimens for the analysis. Both setting seemed to be well planned and well written. The results of these experiments led the authors to validate the results implementing visual models to compute the contrast change in the pupil of a scorpionfish as perceived by an untreated triplefin when producing an ocular spark.

Although the data appear sound, I have a minor concern concerning the interpretation of the results based on the recorded movements, not taking into account any behavior analysis; however, the results support all the conclusions and their hypothesis of ocular spark improving visual detection of cryptic organisms.

Author's Response to Decision Letter for (RSPB-2019-0473.R0)

See Appendix A.

RSPB-2019-2292.R0

Review form: Reviewer 1 (Jolyon Troscianko)

Recommendation

Accept as is

Scientific importance: Is the manuscript an original and important contribution to its field?

Good

General interest: Is the paper of sufficient general interest?

Excellent

Quality of the paper: Is the overall quality of the paper suitable?

Acceptable

Is the length of the paper justified?

Yes

Should the paper be seen by a specialist statistical reviewer?

No

Do you have any concerns about statistical analyses in this paper? If so, please specify them explicitly in your report.

No

It is a condition of publication that authors make their supporting data, code and materials available - either as supplementary material or hosted in an external repository. Please rate, if applicable, the supporting data on the following criteria.

Is it accessible?

Yes

Is it clear?

Yes

Is it adequate?

Yes

Do you have any ethical concerns with this paper?

No

Comments to the Author

I believe the authors have done a good and thorough job of addressing my concerns, and those of the other reviewers. I confess I still have my doubts about the effectiveness of this potential mechanism. The experiments in this study are by no means perfect (given the difficult working conditions and small sample sizes), but given the available data I think the conclusions are justified by the data. So on balance I think this study is acceptable for publication.

Review form: Reviewer 4 (Patricia N. Schneider)

Recommendation

Accept as is

Scientific importance: Is the manuscript an original and important contribution to its field?

Excellent

General interest: Is the paper of sufficient general interest?

Good

Quality of the paper: Is the overall quality of the paper suitable?

Good

Is the length of the paper justified?

Yes

Should the paper be seen by a specialist statistical reviewer?

No

Do you have any concerns about statistical analyses in this paper? If so, please specify them explicitly in your report.

No

It is a condition of publication that authors make their supporting data, code and materials available - either as supplementary material or hosted in an external repository. Please rate, if applicable, the supporting data on the following criteria.

Is it accessible?

Yes

Is it clear?

Yes

Is it adequate?

Yes

Do you have any ethical concerns with this paper?

No

Comments to the Author

The manuscript has been strengthened and clarified by the edited text and the authors have largely addressed my concerns.

Decision letter (RSPB-2019-2292.R0)

17-Dec-2019

Dear Professor Michiels

I am pleased to inform you that your manuscript RSPB-2019-2292 entitled "Redirection of ambient light improves predator detection in a diurnal fish" has been accepted for publication in Proceedings B.

The referee(s) have recommended publication, but also suggest some minor revisions to your manuscript. Therefore, I invite you to respond to the referee(s)' comments and revise your manuscript. Because the schedule for publication is very tight, it is a condition of publication that you submit the revised version of your manuscript within 7 days. If you do not think you will be able to meet this date please let us know.

[http://datadryad.org/submit?journalID=RSPB&manu=\(Document not available\)](http://datadryad.org/submit?journalID=RSPB&manu=(Document%20not%20available)) which will take you to your unique entry in the Dryad repository. If you have already submitted your data to dryad you can make any necessary revisions to your dataset by following the above link. Please see <https://royalsocietypublishing.org/journals/ethics-policies/data-sharing-mining/> for more details.

Sincerely,

Dr Daniel Costa
mailto: proceedingsb@royalsociety.org

Associate Editor
Board Member
Comments to Author:

We have now heard back from two experts in the field. On the basis of their assessments, I am pleased to recommend acceptance. Congratulations on an interesting paper.

Reviewer(s)' Comments to Author:

Referee: 4

Comments to the Author(s).

The manuscript has been strengthened and clarified by the edited text and the authors have largely addressed my concerns.

Referee: 1

Comments to the Author(s).

I believe the authors have done a good and thorough job of addressing my concerns, and those of the other reviewers. I confess I still have my doubts about the effectiveness of this potential mechanism. The experiments in this study are by no means perfect (given the difficult working conditions and small sample sizes), but given the available data I think the conclusions are justified by the data. So on balance I think this study is acceptable for publication.

Decision letter (RSPB-2019-2292.R1)

19-Dec-2019

Dear Professor Michiels

I am pleased to inform you that your manuscript entitled "Redirection of ambient light improves predator detection in a diurnal fish" has been accepted for publication in Proceedings B.

Open Access

Paper charges

Sincerely,

Appendix A

04-Jun-2019

Dear Professor Michiels:

I am writing to inform you that your manuscript RSPB-2019-0473 entitled "Active sensing with light improves predator detection in a diurnal fish" has, in its current form, been rejected for publication in Proceedings B.

This action has been taken on the advice of referees, who have recommended that substantial revisions are necessary. With this in mind we would be happy to consider a resubmission, provided the comments of the referees are fully addressed. However please note that this is not a provisional acceptance.

Sincerely,

Dr Daniel Costa
mailto: proceedingsb@royalsociety.org

Associate Editor
Board Member: 1
Comments to Author:

We have now heard from four reviewers -- and it is clear that there is a split decision with respect to your manuscript. I am recommending rejection but I hope that you will consider re-submitting a revised manuscript that deals with the critical comments of the reviewers.

Thank you for offering us this opportunity.

Reviewer(s)' Comments to Author:

Referee: 1

Comments to the Author(s)

Many animals are easy to spot at night by using a torch and looking for their eye-shine. The idea that fish could be doing the same thing to detect predators and prey is a fascinating prospect. This manuscript presents a large body of work to demonstrate how tripplefin fish could use special reflective irises to create “flashes” or “sparks” required to detect the eye-shine of nearby predators.

I know that this issue is highly controversial in some circles, which is why I’m glad to see the authors have been so thorough in their analysis, and I believe it is an exciting new mode of predator-prey interaction worthy of publication in Proceedings B.

There is no reason in principle why the underlying physics cannot produce eye-shine (at least at some distance), particularly given the incredible range of photonic structures seen in nature. Fish have large eyes which protrude from the sides of their heads. Lenses such as this will focus substantial down-welling light outside of the retina, and onto the iris instead (so there is the collection of lots of light). Then a reflective surface/mirror on the iris can be used to reflect that light. Adding motor control to the mirror will mean it can be pointed in different directions. The authors have performed all of the modelling required to show that this mechanism can work in principle, and have also performed a range of behavioural tests to show that the tripplefin can use the mechanism to avoid predatory fish.

→ *Thank you. We agree that the building blocks for active photolocation are ubiquitous. The challenge is indeed to show under which conditions this mechanism could work. This paper is an important first step in that direction.*

Overall I have no major concerns about the methods or analysis. I have skimmed through the R code to try to work out how the modelling works, and it seems sensible. The stats seem good (though see minor points below).

→ *We appreciate the effort of referee 1 to screen the scripts and statistical analyses for possible mistakes or inaccuracies.*

General Comments:

An alternative hypothesis (which the authors don’t mention) could be that the tripplefin use their sparks to signal to the predators that they can see them (predator signalling and inspection is quite common in fish). Given the tripplefin of all treatments could avoid the predators, what would the predictions of this theory be, and can it be tested with the existing data? This might also lead to expected differences between the fish of different treatments, though it’s not immediately clear to me what those predictions would be, and in any case the ability to avoid the presence of a predator is a more parsimonious explanation for the author’s results.

→ *This alternative hypothesis differs from what we test here. We can therefore only speculate whether some form of "pursuit deterrence" exists in triplefins. In order to test it, we would first need additional information, e.g. are triplefins wearing a shading hat aware that they cannot produce an ocular spark after visually detecting a scorpionfish? Wouldn't this make them more cautious when approaching the predator? If so, we should have obtained the opposite effect of what we observed: shading hat fish staying further away from the predator compared to the controls. Otherwise, if shaded triplefins are unaware of whether an intended ocular spark is "on"*

or not, we would expect all treatments to keep the same distance from a scorpionfish (again assuming they spotted it already).

- *As far as we can tell, pursuit deterrence usually involves specific, whole-body behaviour (e.g. stotting in gazelles) or conspicuous displays or sounds. From our own observations, we consider ocular sparks to be too subtle to fit in this scheme. If triplefins use pursuit deterrence, it is more likely to be e.g. first dorsal fin flicking or bobbing, both of which are shown frequently when they are alerted and excited by something. Given that they involve physical movement of structures bigger than ocular sparks, they may be easier to see by a scorpionfish.*
- *As indicated by the referee, the hypothesis that ocular sparks contribute to visual detection of a cryptic predator is a more parsimonious explanation for our results.*

Regarding the ability of tripplefish of all treatment types to detect scorpionfish, presumably the scorpionfish moved every now & then, so this would be a clear cue for the tripplefish that they are not a rock?

- *This is correct. Scorpionfish did occasionally move. However, scorpionfish normally freeze when approached by a triplefin, presumably to prevent scaring away potential prey. Since both treatments would have been affected in the same way, we would argue that all treatments are affected equally. Our results are probably conservative for this reason. The effect of a shading hat may have been stronger if scorpionfish would not move at all.*
- *We shall use dummy scorpionfish in future experiments to reduce this source of noise.*

Methods: The distance measurement method used is certainly not optimal (i.e. judging distance based on a ruler, which will have a different angle to that being measured). Ideally the researchers would have used cameras to record the positions of the fish at regular intervals, and then distances could be measured from the images (near infrared images would also make it impossible to distinguish between hat treatments). The repeatability analysis (comparing estimates from the two divers) goes some way to addressing this, however we must assume both divers were not naive to the hypothesis being tested. So if there were any measurement bias, all we know is that both divers were equally bias (which is highly unlikely, but possible).

- *We are aware of this, as shown by the repeatability analysis. At the time of these experiments, we decided against cameras mainly because of the expected poor image quality resulting from the unfavourable combination of large tanks and small, cryptic fish. We are, however, working towards automated image analysis in field tanks optimised for this purpose.*
- *There was little scope for subjective judgement: Centimetre markers along both long sides of the tank were used to align the position of the head of a triplefin in between. This is now stated explicitly in the text (lines 164-172).*

How was the retro-reflectance of the scorpionfish eye measured, from how many samples? Wouldn't you need an ophthalmoscope-based setup in a darkroom to measure this? The reflectance used in the code seems to just be the first column in the file (about 5% reflectance at 500nm, which sounds plausible, but isn't labelled). This value is quite critical to the calculations though, so needs much more clarity. I think figure 4 is really valuable here to highlight the parameter-space. e.g. even if the retroreflectance from the scorpionfish pupil is very low, and the spark is within the range measured, the retroreflection should still be visible 5-6cm away. I think the authors should highlight this graph earlier on (with the equations), and maybe specify what the measured parameter ranges were.

- *Eyeshine reflectance measurements were indeed taken in a dark room using an ophthalmoscopic setup, as described in detail in Santon et al. 2018 Scientific Reports DOI:10.1038/s41598-018-25599-y. In this paper, retroreflective eyeshine is termed "SAR narrow-sense eyeshine", the retroreflective component of the stratum argenteum-reflected eyeshine (lines 280-283).*

- *The unnamed columns in the file shows reflectance values for each wavelength (= rows) from the weakest (first column) to the strongest reflectance (last column) based on actual measurements, and interpolated in fixed steps on a linear, "continuous" scale (all the columns in between). These columns have now been named. Furthermore, all data files have been checked and column labelling has been adjusted where it was incomplete or missing.*
- *Please note that we always express reflectance as a proportion (not %) to facilitate calculations. A reflectance of "5" therefore means 5 times as bright as a diffuse white standard under the same illumination and geometry, or 500% (lines 264 and 391-392).*

Specific comments:

The grammar needs sorting out in a few places (particularly the intro).

- *Thank you for the suggestion. Some sections of the manuscript have been rewritten.*

L215 Was a different distribution used because the tank was smaller (making the distance data more bounded and less normal?) edit: you explain this later, maybe say "see below", or just don't mention it here.

- *This statement was indeed not necessary at this point and was deleted. It is now only explained in the next section. Also, a statement was added to explain why the data distribution in the third experiment differed from that one used in the first two experiments (section "Predictors and transformations" lines 248-252).*

L209 If you don't average the control-hat and no-hat fish do you get the same results? Or is the sample size too small? It would at least be good to add this full analysis if possible (e.g. as supplementary data), which should at least show the same trends, even if the sample size is small.

- *We added a new figure (S4) to the electronic supplementary materials in which the data of the first two experiments are shown without pooling the controls (using only the triplets where hats stayed on for the whole experiment). For the first field experiment, this reduces the available sample size from 24 to 13 (north facing tanks) and from 19 to 9 (south facing tanks). For the lab experiment, we only used full triplets. The effect sizes and credible intervals show that all trends are qualitatively similar to those shown for the averaged controls, but that the sample size in the (more noisy) field experiment had become too small to achieve statistical significance.*

I would like more clear drawings/photos/descriptions of the field tanks. I'm slightly confused as to how exactly the control stone and scorpion fish are presented. Edit: I see in the SEM there are nice figures, maybe point towards these in the text.

- *We added more references to the figures found in the supplemental materials.*

L228-230 I'm confused as to why models for north and south oriented tanks were calculated separately given orientation (and its interactions) was added to the model above (L225). Do you mean you re-ran the same single model but specified different contrasts within it, or did you split the data and run two models? Edit: this makes a bit more sense after seeing the different effects in figure 4.

- *As the referee realises, the response in the south facing triplefins was so different that it seems likely that triplefins perceived the scorpionfish differently from this perspective. This is why we decided to do the analysis separately.*
- *The model that compares the response of the controls includes orientation as a factor.*

L280 what is a liquid light guide?

- *Liquid light guides are liquid-filled, flexible tubes and are used in the same way as glass-fibre light guides for spectrometry or illumination. LLGs mix the light better and cannot break. However, the width of their spectral transmission is narrower than that of glass fibres. We use LLGs that have a good coverage in the visible range.*
- *Because all these details are not essential, we now write "liquid-filled light guide". This part has been moved into the ESM for space limitations (ESM line 138).*

L305 Equations 2 & 6 are in the SEM.

Your equations don't make it clear that the calculation is repeated at each 1nm increment based on reflectance spectra before summing to cone-catch quanta (e.g. maybe the equations should have the relevant functions added).

- *Thank you for the suggestion. We corrected the reference to "(sum of the equations explained in ESM figures S3 and S4)" (line 299).*
- *We added a line in the methods specifying that both fluxes were calculated at a 1 nm resolution before calculating cone-catch quanta (line 301-302).*
- *We also added a line to the legends of ESM model displays (figures S3-S4).*

L414. Maybe say "face detection" instead? Face recognition implies identification of an individual based on their face (at least in other spheres of science). Fish can do thi, but I think that's not the point the authors mean.

- *Thank you for pointing out the difference. Changed as suggested (line 452-453).*

Jolyon Troscianko

Referee: 2

Comments to the Author(s)

In this study, the authors test the effect of an ocular reflector on the predator detection abilities of the triplefin in the lab and in the field using a predator (scorpionfish) or a predator mimic (rock). I have a number of major concerns regarding the concepts explored in this paper as well as the methodologies used to test these concepts.

The authors use the concept and term "active sensing" incorrectly throughout this manuscript. Active sensing is the use of self-generated energy to sense the environment (Nelson and Maclver, 2006). Examples of active sensing that the authors present are the light organs of lantern fish and flashlight fish where these animals produce the luminescence by which they hunt. These examples are not comparable to the light reflected from the iris in triplefins. Light reflected from the iris is passive, even if the animals can sense it with photoreceptors in or near their eyes.

- *We agree with referee 2 that there is a definition issue, caused by a novel phenomenon not previously considered by the definitions used by Nelson & Maclver (2006).*
- *In our view, "active" refers to the **behavioural control** of light redirection (Michiels et al. 2018). Behavioural control is also the basis for "contact active sensing", e.g. insect antennae or rodent whiskers (Nelson & Maclver 2006). Yet, we also see an element of "teleceptive active sensing" (sensing reflections of self-generated signals), but with the restriction that diurnal fish do not*

produce light, but behaviourally redirect ambient light. Given that it improves perception by changing the properties of a target, we think this process should not be described as regular vision, but deserves a place in the realm of active sensing mechanisms.

→ *We changed the title of the paper by removing "active sensing" and explain the terms "teleceptive" and "contact" active sensing in the introduction (lines 63-68).*

The no-hat and clear-hat controls should not be treated as the same group. They are different types of controls.

→ *This is correct. We designed both controls to provide information about different aspects of the manipulation. Whereas the unhatted sham treatment allowed us to assess whether the presence of a hat affects behaviour, clear hats represented the "no shade" control for opaque hats (now explicit in lines 121-123).*

→ *The decision to pool the controls (by averaging) was not planned a priori, but was taken to deal with data loss caused by hat loss in the field trials, once we assured that a clear hat did not significantly change fish response compared to the unhatted sham-treatment (see statistical analyses, ESM tables S1-S2 and ESM figure S2).*

→ *As already mentioned above (replies to reviewer 1), we added a new ESM figure (S2) to the electronic supplementary materials in which the data of the first two experiments (lab and first field experiment) are shown without pooling the controls (using only the triplets where the hats stayed on for the whole experiment). These new figures illustrate that both controls do indeed respond very similarly and that pooling them does not generate new effects (compare with figure 2). In short: attaching a hat to a triplefin had no significant effect, as long as it was clear.*

→ *Averaging controls almost doubled the sample size available for analysis in the first field experiment from $N = 13$ to $N = 24$ in north facing tanks and from 9 to 19 in south facing tanks. This strongly improved the mean estimates (smaller error terms). It did not qualitatively change the direction of a trend already present (figure S2).*

→ *The laboratory data are unaffected because sample size stayed the same. This consistency in the results (compare figure 2a with ESM figure S2a) between the two different approaches (pooling the controls or not) shows that they are qualitatively similar. We decided to also average the controls in the analysis presented for lab experiment for the sake of consistency.*

→ *The section "Statistical model choice and pooling of controls" now provides more background for this decision (lines 209-226).*

Another major concern is the use of the concept "diurnal active photolocation". The authors treat "regular vision" as a static and rigid sense instead of the dynamic ability to detect light under many environmental conditions and integrate that information with other cues (such as predator chemical cues). The ability to detect a predator visually is part of vision, and the use of vision to interact with the environment is a whole field in and of itself (visual ecology). Many visual systems are adapted to such processes. Vision, plus the ability to detect the chemical cues of the predator, make this a system where the animals can rely on the integration of their sensory systems to evade a predator.

→ *We are not sure how to improve the ms based on this very general statement. We consider vision neither static nor rigid. But it is correct that we focus on a limited number of key parameters (while controlling for confounding factors). A reductionist approach lies at the core of every experiment. In addition to providing clearer answers to a specific question, it also assures that a reasonable sample size per treatment combination can be obtained. It also facilitates replication by others. We are realistic about these limitations: Each new result is the starting point of a new hypothesis and experiment with other or more parameters of interest.*

→ *We are of course well-aware of other senses. This is why stones were visually presented in the presence of an invisible scorpionfish and vice versa, because we assumed scorpionfish can perhaps be smelled, and they are noisy too (Bolgan et al. 2019 J. Exp. Biol.).*

→ *We agree with this referee that this is only a start. As always, there is need for further experiments to gradually produce an integrative, comprehensive understanding of the visual interaction between a prey fish and its predator. This, however, is well beyond the scope of a single paper that already reports data from three independent experiments and a model calculation.*

Perhaps changing the light that is entering the eye by placing either the shaded or unshaded hat changes the visual environment simply by reducing the amount of light that reaches the eye. Fish eyes are designed to reflect light so that superfluous sunlight cannot reach the photoreceptors in order to reduce the signal-to-noise ratio (Land and Nilsson, 2012). Thus, by changing the amount of light entering the eye, this changes the perception of the environment in ways that may alter resolution and/or sensitivity. This may result in poorer vision and the lack of ability to properly detect a predator visually, causing the prey to move closer to detect the predator by other modes of sensation such as chemosensation.

→ *We do not know to what extent visual perception in shaded fish was affected. If such an effect is present, however, blocking downwelling light should reduce blinding sunlight (e.g. when humans wear a cap in the sun), improving vision rather than deteriorating it. This follows logically from the argument provided by the referee. Furthermore, as long as the interaction takes place in a daytime environment, it is safe to assume that all our tests involve photopic vision.*

→ *Shading hats may affect vision somewhere above the horizontal plane. However, we expect it to be minor for the paradigm we exposed fish to in our experiments: Hats are small, with the two wings raised well above the head (figure 1). They allow free eye and head movement. Minor tilting of the body or head is sufficient for an opaque-hatted fish to look up, which they do by propping up themselves with their pectoral fins.*

→ *The experiments only show that the two hat types behave accordingly to what was predicted by our working hypothesis. They do not exclude mechanisms for predator detection other than active photolocation. After all, shading hatted fish also responded to scorpionfish (as shown by the shorter distance they kept from a stone). This is why the visual model is important: It indicates that the empirical data **can** be explained by diurnal active photolocation in unhatted individuals.*

The authors fail to mention that triplefins have at least four chromatophore patches on their iris. They refer only to the most ventral patch. Is only the patch below the lens useful? What about the breeding male phenotype that has no patch? Are they less able to detect predators? This would be interesting to test given the polymorphic nature of these males.

→ *T. delaisi **can** have up to four chromatophore spots. An (unpublished) sample of pictures of eyes of 248 T. delaisi individuals showed that 4 (1.6%) showed no spot. 197 (79.4 %) only showed one, single ventral spot. Two ventral spots were seen in 13 individuals (5.2%). A ventral and a caudal spot was seen in 27 (10.9%). All other combinations were rare (total 2.8%). All four chromatophore positions (dorsal, caudal, ventral, rostral) were represented in the sample, but never combined in a single individual (max was 3, in 2 individuals). If more than one patch is present, the ventral patch is always the largest. Fish sitting in a regular, horizontal posture, can only focus light on this ventral patch. Since triplefins can sit on surfaces of any orientation, we have observed vertically positioned fish that focused an ocular spark on the smaller frontal or caudal spot of the iris (head down or up). We agree with the referee that they could potentially be used in nature. However, in our experiments all distances were measured in fish sitting on a horizontal substrate. Summarizing, we think that the contribution of ocular sparks generated on other chromatophore spots than the ventral one can be safely ignored.*

→ *Breeding males defend an upside-down territory on a heavily shaded roof of a crevice or overhanging rock with no direct sunshine (and no scorpionfish). Moreover, they are preoccupied with attracting and courting females, fighting competitors, spawning, and defending eggs.*

Catching a male in this phase leads to an almost instant colour change back to an intermediate colouration with a grey head. We have always been intrigued by the question raised by this referee ourselves, but for now we have ignored breeding males.

Why do animals need to acclimate to visual stimuli for 14+ hours? I suspect that the long acclimation period allowed animals to acclimate to the chemical cue of the predator, rather than acclimate to any visual stimulus. Animals should acclimate to visual stimuli quickly, if this is the only sensory cue that they are using.

- *+14 hours of acclimation included one night, during which triplefins are inactive. Actual daytime acclimation was in the range of ~ 6 hours. This is now mentioned in the ms (lines 151-154).*
- *Long acclimation was a cautious decision taken initially for the first two experiments (laboratory and first field experiment). We wanted to be sure that hatted fish obtained enough time to fully recover from the hatting procedure and had time to explore their new environment. Moreover, fish were tested in long tanks (~ 1 m), while they themselves are only 3-4 cm and move around slowly (see Methods). This explains why we released fish in the tanks the evening before observations started.*
- *We later learned by experience that full recovery from anaesthesia is achieved within 3 h, allowing us to hat and observe fish on a single day. We also learned from the first two experiments that shorter tanks (50 cm) are sufficient to detect a response. Furthermore, pilot experiments showed that individuals move around more on dark sand, resulting in a faster response to a stimulus. Consequently, hatting and observation can take place on the same day, largely eliminating the problem of hat loss during ~48 h in the two first experiments. All these improvements were implemented for the first time in experiment 3 (field 10 m).*
- *These explanations have been added to the methods (lines 174-196).*

I also find the timing of data collection for the lab experiments to be odd. There is so much unaccounted time that the animals are not being observed where they may be moving around, exploring, or interacting. What is the significance of recording their positions for 5 minutes every 2-3 hours? It is quite understandable to have field experiments be timed as they are since bottom-time is restrictive for data collection in this experiment.

- *Point observations are a common form of subsampling in behavioural biology. They allowed us to run several independent tanks in parallel while collecting raw data in real time, making it a very efficient way of data collection within the limited bottom-time available to SCUBA divers. Given that triplefins do not move around most of the time (lines 104-106), reduced temporal resolution in favour of larger sample size seemed a good compromise at first.*
- *Moreover, we had no prior experience with the response of triplefins to the visual stimuli. Hence, we cautiously decided in favour of a longer-than-needed time period.*
- *Small size and cryptic colouration made it impractical use video recording in the long tanks (~ 1 m) used in experiments 1 (lab) and 2 (first field experiment), for these durations.*
- *We learned-by-doing as can be seen in the 3rd experiment (2nd field experiment), where we used a higher temporal resolution and a shorter time window. This turned out to be sufficient and informative. This is a concept that is further developed to include video-recording.*

Referee: 3

Comments to the Author(s)

This paper examines the exciting hypothesis that some fish may use an ocular spark to improve their detection of predators. Using triplefin fish as an example, the authors use lab, field, and visual modelling methods to investigate this hypothesis. By making triplefins unable to produce optic

sparks through shaded hats, the authors examine if hatted fish show a decreased ability to respond to a camouflaged predator (via smaller approach distances). Although I find the idea and potential mechanism exciting, in my opinion both the visual modelling and behavioral data have issues (some with interpretation), and as a result do not fully support the claims made in this paper.

Although the paper presents the behavioral data first, I find it easier to start with the visual modelling. The visual modelling section found that “ocular sparks did not generate chromatic contrast above discriminability threshold” and that achromatic contrasts were only exceeded at values smaller than 6-8 cm depending on the location. In many ways, these values represent best case scenarios. For example, 1) the Michelson contrast threshold of 0.008 used in the modelling is the most sensitive found in a previous study and may be less sensitive depending of the angle of the stimulus [cited ref 13], 2) the radiance of the ocular spark used was “the highest value for each fish relative to the standard”, and 3) the ocular spark was only measured at 45° despite possibly (likely?) having angular dependence.

- *We considered the suggestion of the reviewer, yet we prefer not to present the model first because it is based on spectrometric measurements from the second field experiment.*
- *We value experimental results more - even if they are noisy and harder to obtain. In this study, the model is a post-hoc addition that helps assessing whether the empirical data can be explained by diurnal active photolocation using an ocular spark.*
- *It is true that the 0.008 Michelson contrast was the most sensitive found. Yet, this value is not very different from values obtained for two freshwater fish and it is the first ever to be determined for a marine fish species. This makes it difficult to assess whether this an unusually low value, or perhaps the rule for small, benthic fish.*
- *Measuring ocular sparks in live fish is not straightforward. Fish movement (eyes, head, breathing) resulted in microscopic, uncontrollable movements during each spark measurement. Picking the highest value for each eye was a logical consequence of the fact that some measurements were clearly off (e.g. inclusion of a signal from the nearby reddish parts of the iris). Hence, the "highest" value only means "closest" to the actual performance of an ocular spark. It is important to see this in its wider context: These values are **only** used to define an appropriate width of the scale in the modelling (figure 4). Note that the lower end of the scale starts at < 80% reflectance relative to a diffuse white standard (expressed as a proportion, 0.8). This is a conservative start for the reflectance range of a spot of focused light.*
- *Ocular sparks show little if any angular dependence in the horizontal plane, and can therefore be considered diffuse reflectors in that plane. See Bitton et al. 2019 Sci. Rep. for details.*
- *In conclusion, we have not tailored the model to our expectations. This is also illustrated by the fact that we pay more attention to the average results of the model rather than the extreme, best results. We **underestimate** scorpionfish retroreflection (one of the most important parameters in the models) as the measurements did not consider whether it was looking at and focusing on the source, which strongly increases the reflectance values (Santon et al., 2018).*

Unfortunately, the authors do not report what % of photons are redirected via the optical spark, nor what % of photons that hit a scorpionfish pupil and iris is actually reflected. I tried to find this information in the supplementary tables, but am uncertain if I am interpreting it correctly because the columns of ESM13 and ESM15 are not labeled (I'm assuming its for different individuals, but may be wrong). Reporting these values would go a long way in understanding just how subtle this light is in a natural condition. That chromatic contrast was not changed (despite the fascinating pictures in Figure 1 d-f) and that the achromatic contrast threshold is only reached at very short distances suggests that this light production may be too subtle to be functional.

- *All data were and are provided with this submission, and we apologise for any lack of clarity. It may have been confusing that we used proportions and not percentages for e.g. reflectance values. Some data were indeed poorly labelled. We took care of all of this with this submission.*

- *The "missing" data were in ESM 13, 15 and 16 (now part of the Drayd submission with the same file numbers). Each column (except for the first two) of each file shows the proportion of light reflected across a wavelength range from the minimum (column 2) to the maximum measured (column 40) on an interpolated scale. This is now clearly specified.*
- *Ocular spark reflectance expresses the radiance of an ocular spark relative to that of a diffuse white reflectance standard exposed to the same downwelling light and is expressed as a proportion (term "S" in ESM figure S4).*
- *Retroreflectance in scorpionfish is also expressed as the proportion of photons returned to the source relative to a diffuse white reflectance standard in the same position, in an otherwise dark room. The values are much higher than those of an ocular spark: Scorpionfish pupils are retroreflectors, whereas the chromatophore patch on which ocular sparks are reflected seem to act as a diffuse reflector.*
- *The referee overlooks the fact that the number of photons reflected by a diffuse source and striking a scorpionfish pupil increases exponentially as the distance between the two becomes shorter. We placed two demos online to illustrate that (1) indirect, diffuse sources can indeed reflect enough light to illuminate nearby surfaces (DOI: 10.13140/RG.2.2.20239.64163) and that (2) retroreflection works remarkably well for weak sources (DOI: 10.13140/RG.2.2.10168.19208). These demos do not prove the validity of our biological hypothesis. Yet, they show the counterintuitive small-world effects that arise when a diffuse coaxial source and a retroreflective target interact.*

With the visual modelling suggesting a limited – although feasible --- range of function, one would like strong behavioral data to support it. The modelling suggests that if there is any predator avoidance effect from the optic spark that it should occur only at very short distances. As such I'd love to minimum approach distance to the predator, but this data was not collected, and just average distance is shown instead. This makes it hard to tell if the difference between treatments (when present) is occurring at a similar distance to what is predicted via visual modeling.

- *We fully agree that "shortest approach" data are highly informative. It is also true that this information cannot be extracted from experiments 1 and 2 due to our deliberate preference for a delayed, next-day start of data collection and the low temporal resolution, as explained earlier.*
- *However, experiment three (2nd field experiment) did exactly what the referee is asking for. Although the data were not collected continuously, the instantaneous start and the much higher sampling rate allowed to infer minimal approach patterns in raw distance measurements. A considerable number of fishes of both treatments visually inspected the stimulus at distances where active photolocation can work - as indicated by the modelling results.*
- *The figure 3 shows actual (raw) distances across time, not averages.*
- *Active photolocation is not the only way by which triplefins detect scorpionfish. This is illustrated by the fact that shading-hatted fish kept a longer distance from a scorpionfish than from a stone. As can be read in the manuscript, active photolocation should be seen as a supplemental, last-resort mechanism that makes a difference when regular vision fails at detecting a particularly well-camouflaged scorpionfish, as explained in the discussion.*

Video analysis of the initial approach to the predator would be much more convincing than locations at 5 spaced times over the course of a day. Without this data I feel uncomfortable attributing the differences to the optic spark as there could be various other factors involved (including those suggested by the authors such as the range reduction from wearing a hat, or the impossibility to blind the experiment). There may be some way to pull out this data from Figure 3 (or at least show immediate minimum distance) but as currently presented it is not there.

- *Small size and cryptic colouration made it impossible to record fish on video in the long tanks (~ 1 m) used in experiments 1 (lab) and 2 (first field experiment). Moreover, the time span was too long for battery-driven video recording in the field.*
- *In the 3rd experiment (2nd field experiment) we used 50 cm tanks and a shorter time window, which turned out to be sufficient and informative. This experiment represents a paradigm that is suitable for video-recording in the future.*
- *Minimal distances can easily be read directly from figure 3 and represent what the referee is asking for - albeit at a lower temporal resolution.*

Additionally, I have a few statistical questions (I do have experience with these tests, but am not an expert). I'm used to papers that use GLMM or LMM use model fitting to pull out the most parsimonious models, but this has not been done (or at least is not present in the paper/supplementary materials or I missed it).

- *Model selection has been implemented, as (was and is) described in the section "Triplet as random factor and model selection" in the methods. It is based on AIC scores, which is a standard procedure.*

I'm also uncomfortable that the authors "averaged the data of the two control-treated fish per triplet" because there was no significant difference between them. Just because a difference is not significant doesn't mean that there isn't an effect. Considering all the other factors influenced by wearing a hat, I'd love to see the data presented in a clear fashion where I can compare between the transparent and opaque hats, but this is not possible.

- *This comment was also made by referees 1 and 2. Our answer here overlaps with what we replied earlier:*
 - *The controls indeed provide different information: Whereas the unhatted sham treatment allowed us to assess to what extent the presence of a hat affects behaviour, clear hats represent the "no shade" control for opaque hats.*
 - *The decision to pool the controls (by averaging) was not planned a priori, but was taken to deal with data loss caused by hat loss in the field trials. It was done after assuring that a clear hatted fish did not influence behaviours significantly compared to the unhatted sham-treatment.*
 - *We now added a new figure (S2) to the Electronic Supplementary Material in which the data of the first two experiments (lab and first field experiment) are shown without pooling the controls (using only the triplets where the hats stayed on for the whole experiment). It illustrates that both controls do indeed respond very similarly and that pooling them does not generate new effects. In short: attaching a hat to a triplefin has no significant effect, as long as it is clear.*
 - *Averaging controls almost doubled the sample size available for analysis in the first field experiment (from N = 13 to N = 24 in the north facing tanks, 9 to 19 for the other orientation) and consequently reduced noise in the mean estimates (shorter error flags). It did not qualitatively change the direction of an already visible effect, as can be seen in ESM figure S2.*
 - *The laboratory data are instead unaffected because sample size stayed the same. This consistency in the results (compare figure 2a with figure S4a) between the two different approaches (pooling or not the controls) shows that they are qualitatively similar. We decided to also average the controls in the analysis presented for lab experiment for the sake of consistency in data analysis.*
 - *The section "Statistical model choice and pooling of controls" now provides more background for this decision.*

Lastly, I find it weird that the authors present all their data as interactions (Treatment x Stimulus). Because the stone data did not vary significantly in any treatment it seems appropriate to present just the scorpionfish data without getting into the more complicated interactions. This is especially relevant for Figure 2b, where it appears the statistically significant difference in the Stimulus x Treatment interaction is driven in part by a slight (likely random) preference for the shading hat fish to be farther away from a stone. The scorpionfish data does not look significant by itself.

- *Given that we did not expect to see a difference between controls and shading hat for "stone", but we did for "scorpionfish", the interaction term is an integral part of our hypothesis. Hence, it has been considered in all models that include the two stimuli, stone and scorpionfish.*
- *The reviewer is correct in observing that the significant interaction in figure 2b is caused by the different direction of the distance effect between treatments when facing a stone or a scorpionfish. However, subsequent models testing only for effects within a stimulus (as suggested by the reviewer) show that a significant difference between the hat treatments is present only in fish facing a scorpionfish. These effects were already indicated as such in the original graphs (now made clearer) and were mentioned in the text too.*

Considering these factors all together, in my opinion the data presented here does not do enough to back the claims of the paper. Figure 2C gives a negative result, Figure 2B appears to be driven to significance by stone differences (also likely some outlier fish never approaching?), and none of the figures present information about the distance at which this behavioral choice is made.

- *We never predicted that diurnal active photolocation works under all conditions. Not finding an effect under certain conditions is therefore not a reason to doubt the overall validity of a mechanism, but highlights that this mechanism has limitations, as do most if not all animal traits.*
- *This work should therefore be assessed as a package:*
 - *(1) **strong effect** in a laboratory setting*
 - *(2) **confirmed in one of two** orientations in a field replicate of (1)*
 - *(3) **close approach effect confirmed** in a second field experiment (early data acquisition, shorter tanks)*
 - *(4) **visual model confirming** the plausibility based on realistic parameter values*
 - *(5) **limitations** of the mechanism*
 - *triplefins can also recognise scorpionfish without ocular sparks (shading hats)*
 - *orientation may matter (yes for 1st field experiment, no for 2nd field experiment)*
 - *one section of the modelled parameter space predicts poor support of predator detection by diurnal active photolocation*
- *We rewrote the start of the discussion to make this clearer.*

Overall this idea is promising and biologically exciting. However I'd like to see 1) reporting of the % of photons produced via the optical spark and the % reflected by the scorpionfish pupil, 2) having behavioral trials that show the distance of predator avoidance roughly aligning with the visual modelling, and 3) stronger overall behavioral data.

- *% Photons was already provided, but as proportions. The descriptions have now made this clearer. We apologize for not making this clear in the first place.*
- *Experiment 3 shows approach distances for clear-hatted fish that overlap with the predictions made by the model for unhatted fish. See red dashed line in figure 3.*
- *Stronger behavioural data: This paper is unique in that a clear significant result from the lab has been tested again, twice, under field conditions, where many sources of additional noise could not be controlled. Yet, the same trends were observed, albeit not in all conditions - which merely show the real-world limitations of the system (see previous replies).*

→ *A posteriori statements of what else we "should have" done are a standard part of the discussion following every experimental paper. We agree that there is always more to do.*

Minor Comments:

Introduction: Paragraphs! I feel the introduction also needs a lot more context. Lines 38-56 are the only ones currently providing this, the rest just describe the experiment. I'd also suggest defining more words to reach a broader audience (retroreflective, coaxial, etc.)

→ *We divided the introduction in three paragraphs.*

→ *In the current version of the introduction, the description of the experimental design was shortened, offering more space for context.*

→ *We have paid attention to reduce the use of technical terms, or to define them better.*

Ln 41-43 A figure showing the proposed mechanism would help your reader a lot here.

→ *The mechanism has been described extensively by Michiels et al. (2018) and can be seen in ESM figure S4. We added a reference to this figure in the text.*

Discussion

Ln 385 Proof of principle seems to strong here

→ *We changed this to "proof of concept" which is defined as "a realization of a certain method or idea in order to demonstrate its feasibility, or a demonstration in principle with the aim of verifying that some concept or theory has practical potential. A proof of concept is usually small and may or may not be complete."*

Ln 390-392 If the mechanism only works when already in range of the scorpionfish, is it really biologically relevant?

→ *At the average (!) detection distance of 7 cm, triplefins have a very good chance to escape a strike by a scorpionfish. For longer distances (more optimal, but still realistic) the mortality risk is probably close to zero.*

→ *We have witnessed triplefins escape a scorpionfish strike even at closer distances,. Triplefins have a remarkable escape reflex and dart away with "lightning" speed. Hence, scorpionfish require triplefins to approach very closely to their mouth to stand a chance.*

→ *The modelled distances are those to the eye of the scorpionfish, not to its mouth. We have no a priori reason to assume that triplefins usually approach scorpionfish frontally. Instead, we assume triplefins move towards scorpionfish from any direction. A triplefin approaching from the side can come close to a scorpionfish's eye, while still being outside the risk zone near the mouth. Although scorpionfish sometimes turn around to strike at a triplefin sitting e.g. near their tail, this takes more time than what the triplefin needs to respond.*

→ *We are aware of the importance of such observations and will collect more data of this kind in the near future.*

Ln 393 I'd love to see more discussion of the limitations to triplefins before going into other species later in this paragraph. What are the limitations? The angular distribution of light? Depths where it would be effective? Etc.

→ *The limitations to triplefins are addressed in the first paragraph. The angular distribution of light redirected by an ocular spark was recently published elsewhere (Bitton et al. 2019 Sci. Rep.). That paper also addresses the factors that affect diurnal active photolocation.*

→ *It is common practice to address wider consequences in the discussion. For us, these are:*

(1) building blocks for active photolocation are ubiquitous,

(2) limitations to triplefins (e.g. seemingly "weak" light redirection) may be less stringent in non-cryptic species, and,

(3) counteradaptations to eye detection are widespread among cryptic piscivorous fish: perhaps

active photolocation has played a role in shaping them.

None of these are controversial or highly speculative, but fit well in a regular scientific debate.

Referee: 4

Comments to the Author(s)

Comments:

This manuscript describes interesting new data on the functionality of diurnal active photolocation. The authors had previously shown that the benthic triplefin, *Tripterygion delaisi*, controlled redirection of downwelling sunlight using their iris, generating a phenomenon called "ocular spark". In this work, they hypothesize that ocular sparks improve visual detection of nearby cryptic organisms, a process they called "diurnal active photolocation". To test this hypothesis, these authors experimentally suppressed light redirection using the benthic triplefin *Tripterygion delaisi* as a model. They manipulated light redirection through an unusual approach, using shaded hats glued on the fish heads. The authors presented results from the effect of hat treatment on the average distance from the stimulus (stone or scorpionfish), immediate responses from the predator scorpionfish, and color difference metrics/chromatic contrasts are measured as just-noticeable differences (JNDs).

On the first experiment, the authors performed a very careful and controlled experiment, to set the stage for the second (field) experiment, a more robust approach, using more specimens for the analysis. Both setting seemed to be well planned and well written. The results of these experiments led the authors to validate the results implementing visual models to compute the contrast change in the pupil of a scorpionfish as perceived by an untreated triplefin when producing an ocular spark.

Although the data appear sound, I have a minor concern concerning the interpretation of the results based on the recorded movements, not taking into account any behavior analysis; however, the results support all the conclusions and their hypothesis of ocular spark improving visual detection of cryptic organisms.

→ *Thank you. The datasets presented here do indeed not include a detailed behavioural analysis. Instead, we only analysed positions and indirectly infer behaviour from them. This simplified form of "point sampling", however, allowed us to collect a meaningful sample size under novel and challenging conditions. Future work will focus more on the behavioural details and use video-recording in tanks designed for optimal imaging.*